# Hydrodynamic Impacts on the Fate of Polychlorinated Biphenyl 153 in the Marine Environment

**Elena Mikheeva** [1,*], **Johannes Bieser** [1] **and Corinna Schrum** [1,2]

1 Helmholtz-Zentrum Hereon, Institute of Coastal Research, Max-Planck-Str. 1, 21502 Geesthacht, Germany
2 Institute for Marine Sciences, Universität Hamburg, Mittelweg 177, 20146 Hamburg, Germany
* Correspondence: elena.mikheeva@hereon.de

**Abstract:** Due to their long half-life, polychlorinated biphenyls (PCBs) tend to contaminate not only coastal areas, but they travel over long distances, eventually reaching remote areas such the Arctic. The physical and biogeochemical features of every coastal area govern the main distribution patterns of freshly introduced PCBs into the marine system. Some of these processes are determined by chemical properties of the individual PCB congener. Thus, atmospheric influx along with ad/absorption on non-living organic material, photolytical and biological degradation processes vary from one PCB congener to another. For a detailed fate analysis of individual congeners, we developed a new chemical model, based on the GOTM-ECOSMO-FABM model framework. Here, we exemplarily present results for $PCB_{153}$ based on 1D simulations of four regions in the North-Baltic Sea. The study area is characterized by different hydrodynamic and biogeochemical conditions. We investigate the impact of resuspension, mixing and the biological pump, sea ice and tides on the final phasal distribution of $PCB_{153}$. Different combinations of these factors lead to the development of different areas of $PCB_{153}$ accumulation, with the formation of hotspot areas, and influence the total uptake and concentration of $PCB_{153}$ in the water column. As a result, two major dynamics determine the fate of $PCB_{153}$ in the coastal ocean: (i) Primary production leads to $PCB_{153}$ being adsorbed by organic material. Partitioning to organic material and sedimentation of organic particles removes dissolved $PCB_{153}$ from the surface ocean and increases atmospheric influx. (ii) Tidal-induced resuspension and mixing control the benthic–pelagic exchange of $PCB_{153}$ and its distribution in the water column. Depending on the resuspension regime and stratification, sediments can become a permanent (Gotland Deep, the Baltic Sea) or seasonal sink for $PCB_{153}$. In regions with seasonal stratification and high near bottom turbulence (Northern North Sea), resuspension events can lead to pronounced peaks in $PCB_{153}$ concentrations and can therefore have a major impact on bioaccumulation. Under the conditions of permanent mixing and high bottom turbulence (Southern North Sea, Bothnian Bay), pollutants are hardly accumulating in sediments.

**Keywords:** PCB; POPs; North and Baltic Sea; 1D model; pollutant modeling

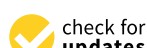



## 1. Introduction

Polychlorinated biphenyls (PCBs) are persistent organic pollutants (POPs) as defined by the UN Stockholm Convention (SC). Due to their physical and chemical properties, they have a tendency to accumulate in different environmental compartments. Although PCBs are banned for production in most countries [1], they are still present in the environment. This is not only due to their long half-life time, but also because of legacy sources, such as improperly maintained organized electronics and manufactural landfills, the illegal discharge of old PCBs, including transformer fluids or leaks from thereof, and the disposal of PCB-containing consumer products into municipal or other landfills not designed to handle hazardous waste [2]. That makes PCBs potentially harmful for biological communities even nowadays [2,3]. In shelf waters, the amount of recently released ('fresh') PCBs is higher than in the open ocean, due to their vicinity to land-based sources. However,

these pollutants are transported toward the polar regions via the open ocean, where they are persistent over long-time spans. This process is known as long-range transport (LRT). The lack of organic matter and sources of degradation allow PCBs to stay in the water column and drift over long distances. That leads to PCB contamination, even in remote areas such as the Arctic Ocean. Coastal waters can be considered the first step of the PCBs journey from land to the open ocean [4–7]. Detailed understanding of their behavior in these regions can help to predict their fate in the environment under changing external conditions (e.g., emissions and climate change).

PCBs are hydrophobic pollutants. They are able to exist in a freely dissolved form; however, they exhibit a strong affinity to bind to organic matter (OM) [5,8]. This phenomenon is defined by the low water solubility of PCBs and their high affinity to organic carbon matrices [4]. There are two different states of dead OM in marine environment—particulate (POM) and dissolved (DOM) organic matter. PCB adsorbed onto POM ($PCB_{POM}$) is effectively transported downwards due to the sinking velocity of POC particles, leading to an accumulation of PCB in sediments. Moreover, PCBs are able to build strong bonds with DOM molecules. This kind of OM is not heavy or dense enough to sink. It stays in the dissolved phase and absorbs dissolved $PCB_{free}$. Finally, freely dissolved PCB and $PCB_{DOM}$ are the most available for direct biological uptake, as they diffuse through the cell membrane and thus bioconcentrate inside phytoplankton [4,9]. PCBs attached to detritus are not available for direct diffusive bioconcentration due to particle size; however, they can accumulate in higher trophic levels via feeding.

These processes are general for all PCBs; however, specific properties of pollutants define the distribution of PCBs between all three phases ($PCB_{POM}$, $PCB_{DOM}$ and $PCB_{free}$). The ability of PCB to bond with OM is described by the octanol–water partitioning coefficient $K_{ow}$, which varies from one PCB congener to another ($logK_{ow}$ is from 4.66 ($CB_1$) to 8.20 ($PCB_{209}$)) [10,11]. Moreover, differences between chemical and physical features of PCBs lead to different transport and degradation patterns, especially in the aquatic environment [12,13]. Chlorinated biphenyls (CBs) have two directly connected phenyl rings with chlorine atoms, substitutional hydrogen in ortho, meta and/or para positions (Figure 1). There exists a total of 209 congeners of CBs, and their properties are mainly determined by their chemical structure. As an example, photo degradation mainly affects higher chlorinated congeners, with the following dechlorination (losing chlorine atoms under UV irradiation) [14,15]. On the other hand, for aerobic biological degradation, a molecule of PCB has to have a specific structure, with two carbon atoms free of chlorine in ortho and meta position at the same side of the benzene ring (Figure 1) [16,17]. Otherwise, bacteria (microbial community) are not able to break the carbon–carbon bonds.

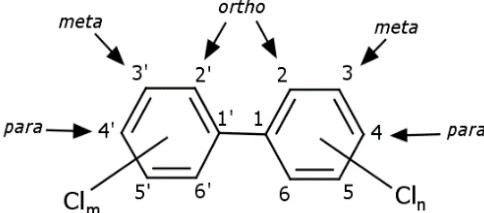

**Figure 1.** General chemical structure of chlorinated biphenyls.

Finally, atmospheric exchange is determined by the Henry's Law constant, which is dependent on the chemical properties of each pollutant (enthalpy and entropy of dissolution) [18].

Thus, potential hot spots of pollutants from the same class vary from one congener to another. The aim of the current research is to estimate PCB hot spots in the marine environment. It is an important task on the way to understand their marine cycling and estimate areas of highest bioconcentration.

These areas differ from one region to another, according to regional hydrodynamic features. Moreover, processes which have a major influence on the flux of OM define the distribution of PCBs in the water column [12]. Regions with a stable stratification are able to keep pollutants in the sediment, once they sink with POM to the bottom. In regions characterized by resuspension, PCBs are constantly resuspended back into the water column. After being resuspended, they are able to become dissolved again due to remineralization processes (secondary source of PCB) [12]. In general, the biological pump strongly controls the production of OM and thus also controls the phasal distribution and transport of PCBs [12].

As there are only limited measurements of a selected PCBs available, it is necessary to employ models to investigate their fate in the marine system. So far, only a few models for PCBs have been published. Mostly, these models are so-called box models, where concentrations of pollutants are calculated as fluxes between different compartments [19–21]. In this case, characteristic hydrodynamic and biogeochemical features of chosen regions are not represented, despite their potential importance for the distribution of PCBs. Along with box models, few studies have also presented 3D models of POPs in the marine environment [12,13], from which the role of environmental conditions in POP fate and transport is evident. These studies usually lack in complexity in terms of the hydrodynamic biogeochemical–pollutant coupling in the water column and dynamic benthic–pelagic coupling. Moreover, the PCB chemistry is neglected or rather simplified in these models—PCBs are typically lumped into a single PCB species instead of using congener-specific physical and chemical properties, despite their importance for degradation and accumulation processes.

The dynamic modeling of individual pollutants transport and transformation in coastal areas can play an integral part of PCBs global fate estimation. However, here, the identified limitations in state-of-the-art models form a significant constraint for the PCB modeling capacities. On one side, the coastal ocean can have great variations of hydrodynamic and biogeochemical conditions on a wide range of time scales. At the other side, the system is dominated by benthic–pelagic feedbacks, which involve frequent changes between processes of sedimentation and the resuspension of organic material [22].

Here, we present a new numerical model for PCB cycling in the marine environment that considers biogeochemical and physical processes in the water column, the benthic–pelagic coupling along with chemical features of the PCB congener. With this model, we investigate different processes in the water column and their influence on the behavior of one PCB congener, considering its specific physico-chemical properties, here—$PCB_{153}$. To isolate the dominant structuring of first order hydrodynamic (and consequent biogeochemical) effects on PCB transformation from horizontal advection and spatial differences in sources, the first simulations were carried out in 1D model setups. We chose four exemplary hydrodynamically different regions in two coastal seas—the North and Baltic Seas. Both North Sea locations are impacted by tides; however, they have different depths and levels of biological production. Gotland Basin is the deepest region in the Baltic Sea, which is characterized by strong stratification. The last area, Bothnian Bay, is affected by advective mixing with seasonal ice cover. These seas are located in Northern Europe, a region of high PCBs production in the past [1,23–25]. Even nowadays, after these pollutants were banned for production by SC [1], they are still present not only in the marine environment, but also in the atmosphere [26].

The developed dynamic model is intended to be universal and able to estimate the fate of all PCBs considering their specific properties. In this manuscript, we present simulations of one congener ($PCB_{153}$).

## 2. Model Description

### 2.1. General Information

The new coupled 1D model, presented in this research, is able to simulate the fate of PCB (or potentially any other hydrophobic POP) in the marine environment, taking into

account not only the hydrophysical properties of a chosen region, but also the chemical–biogeochemical transformations of each pollutant. The main feature of this model is that it can be run on the base of any other marine hydrodynamic and ecosystem model.

The results presented in this article include simulations of one congener—$PCB_{153}$. This pollutant (i) has a very low water solubility and high affinity for partitioning with organic matter, and (ii) has a high tendency for photolytic dechlorination [14,15], (iii) but it is not available for aerobic bacterial degradation due to its structural specifications. $PCB_{153}$ is widespread in the environment, and the necessary data for simulations are available.

For this study, we use the coupled model system to simulate pollutants transformation and distribution in a marine environment. As a driver, we run the 1D hydrodynamic ocean model GOTM [27,28] (Section 2.2) together with the marine ecosystem model ECOSMO [22] (Section 2.3) and couple the PCB chemistry module by using the Framework for Aquatic Biogeochemical Models (FABM) [29] (Sections 2.4 and 2.5) (Figure 2). This model system runs in 1D water column–sediment setups, neglecting horizontal transport. We concentrate on 1D model setups to isolate the dominant structuring hydrodynamic (and consequent biogeochemical) first-order effects on PCB transformation from horizontal advection and spatial differences in sources.

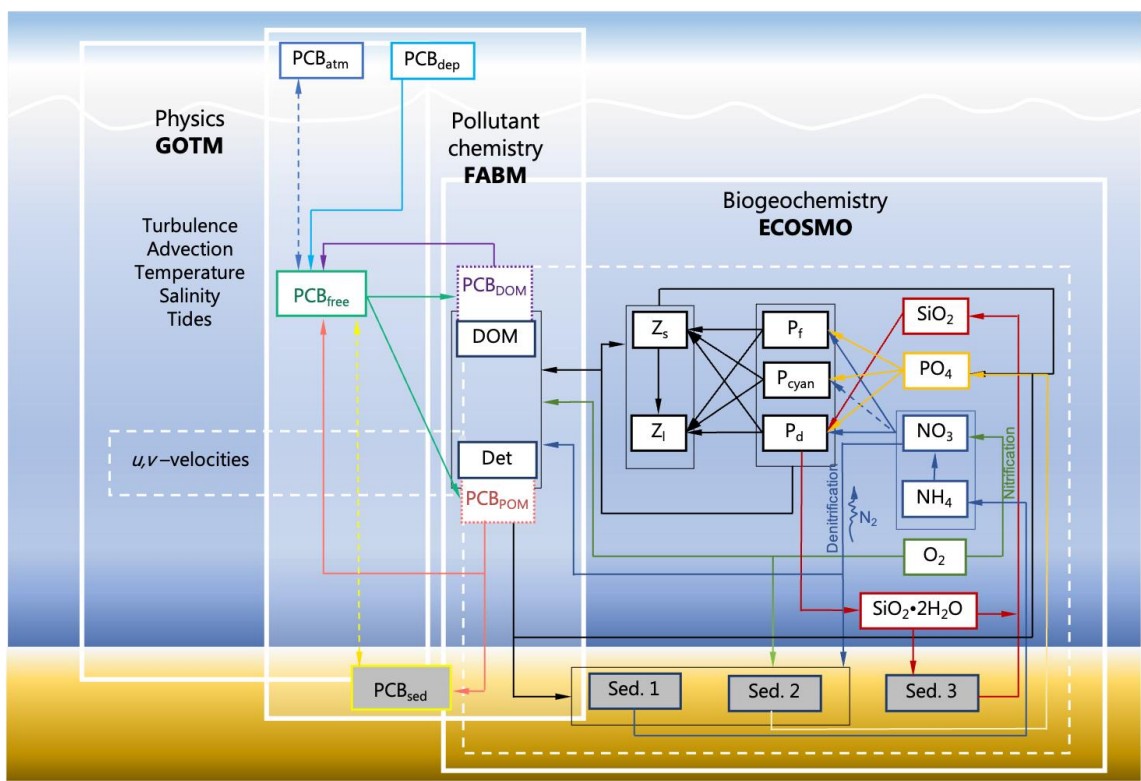

**Figure 2.** Overview of the coupled model GOTM-ECOSMO-FABM (*Det* stands for Detritus) with an emphasis on the flow between the different model compartments of ECOSMO (incl. 2 groups of zooplankton: $Z_s$, $Z_l$; 3 groups of phytoplankton: $P_f$, $P_{cyan}$, $P_d$) (detailed description of ECOSMO in Section 2.3 or Daewel and Schrum, 2013 [22]).

The local change of PCB state variables $C$ is estimated based on prognostic equations generalized in Equation (1) as the sum of transport-related terms (advection as the 1st term, turbulent mixing as a 2nd and settling as 3rd) (Section 2.2). Advection and turbulent mixing are calculated by the hydrodynamic host model (GOTM), and PCB settling is implemented

into the PCB module (FABM). The transformation term $R_c$ includes all chemical processes of PCB (partitioning, chemical reactions, photolysis) (Section 2.5).

$$\frac{\delta C}{\delta t} = -\frac{\delta wC}{\delta z} + \frac{\delta}{\delta z}\left((\nu_t + \nu)\frac{\delta C}{\delta z}\right) - (w_d)\frac{\delta C}{\delta z} + R_c \tag{1}$$

$\nu$, $\nu_t$—molecular and turbulent diffusion coefficient;
$w_d$—sinking velocity (1 m·d$^{-1}$).

At the boundaries, the intercompartmental PCB fluxes between the atmosphere and ocean at the surface and the ocean and sediment at the bottom, are calculated in the PCB module and considered boundary conditions (Section 2.6). The transformation terms are described in detail in Section 2.5. The presented PCB model contains all relevant processes to determine the pollutants environmental fate under different physical and ecological regimes. In the water column, this includes the processes of partitioning (Sections 2.5.1 and 2.5.2) and degradation (Sections 2.5.3 and 2.5.4). At the boundaries, the model considers air–sea exchange and sea–ice interaction (Section 2.6.1), and sedimentation and resuspension (Section 2.6.2). Resuspension from sediment pollutants is considered a pollutant of a secondary source.

Our implementation allows the PCB module to be initialized multiple times representing different congeners with different physical constants and reaction rates. In this case, several models of different congeners can be run in parallel and coupled to each other. That opens an opportunity to include transformation of one PCB into another during further model development (e.g., photolytic transformation of PCB$_{153}$ into PCB$_{28}$).

### 2.2. GOTM—The Hydrodynamic Host Model

Here, we chose to couple the PCB module to a one-dimensional water column model GOTM, as mentioned above [27]. This model is able to represent relevant hydro- and thermodynamic processes related to vertical turbulent mixing in 1D model setup [29].

GOTM calculates the vertical turbulent exchange based on shear and buoyancy production (Equation (2)) [29]:

$$\frac{\delta k}{\delta t} - \frac{\delta}{\delta z}\left(\nu_t \frac{\delta k}{\delta z}\right) = P + B - \epsilon \tag{2}$$

$k$—turbulent kinetic energy;
$\nu_t$—vertical turbulent diffusivity (eddy);
$P$—shear production;
$B$—buoyancy production;
$\epsilon$—rate of dissipation.

The horizontal advection is included via prescribed salinity and temperature profiles either from real-world CTD data or from a three-dimensional hydrodynamic model. At each time step, GOTM forces the vertical structure toward the prescribed profiles using a relaxation time $\tau_R(A) > 0$ (Equation (3)). Salinity and temperature profiles are prescribed from World Ocean Atlas (WOA) data. Simulated profiles are relaxed to the prescribed profiles to account for neglected horizontal advection and lateral sources. Salinity relaxation is implemented according to the following equation (Equation (3)):

$$\frac{\delta A}{\delta t} = \frac{\delta}{\delta z}\left((\nu_t + \nu)\frac{\delta A}{\delta z}\right) - \frac{1}{\tau_R}(A - A_m) \tag{3}$$

$\nu_t$, $\nu$—turbulent and molecular diffusivities;
$\tau_R(A)$—relaxation time scale;
$A$—calculated tracer distribution;
$A_m$—prescribed tracer profile.

The water circulation in the North and Baltic Seas have different time scales, thus adjusted relaxation times for *S* and *T* profiles create a mimic advection simulation, nudging the model toward *S* and *T* observations. Salinity profiles were implemented into the model with two different relaxation times of 86,400 *s* for the North Sea and 432,000 *s* for Baltic Sea (Table 1). Temperature profiles were implemented without relaxation.

**Table 1.** GOTM Specification for current simulations (NNS—Northern North Sea, SNS—Southern North Sea; GB—Gotland Basin; BB—Bothnian Bay).

| | Unit | Value | Comment |
|---|---|---|---|
| | | *General* | |
| Turbulence model | | second-order model | |
| TKE method | | dynamic equation (*k*-epsilon) | |
| Bottom roughness | (m) | 0.03 | The North and Baltic Seas |
| Spin-up | (year) | 10 | Accumulation of pollutants in sediment reaches equilibrium in 10 years |
| Calculation time steps | (s) | 1800 | |
| Relaxation (Temperature) | (s) | - | The North and Baltic Seas |
| Zooming factors ($d_u$ and $d_l$) | - | 1.5; 1.0 | The North and Baltic Seas |
| | | *The North Sea* | |
| Number of layers (the North Sea) | - | 110; 63 | The North Sea: NNS; SNS |
| Relaxation (Salinity) | (s) | 86,400 | The North Sea |
| M2 Tidal Amplitude | (m) | 0.41; 0.7 | The North Sea: NNS; SNS |
| S2 Tidal Amplitude | (m) | 0.16; 0.25 | The North Sea: NNS; SNS |
| M2 Tidal Period | (s) | 44,714 | The North Sea |
| S2 Tidal Period | (s) | 43,200 | The North Sea |
| | | *The Baltic Sea* | |
| Number of layers (the Baltic Sea) | - | 90; 125 | The Baltic Sea: BB; GB |
| Relaxation (Salinity) | (s) | 432,000 | The Baltic Sea |

Vertical grid resolution includes both near-the-surface ($d_u$) and bottom ($d_l$) zooming. As a result, near-surface- and bottom-layer thickness drops by up to 10 cm.

Since simulations were made in 1D, horizontal transport and movement was neglected. However, vertical changes of SSE during tidal activity have an influence on turbulence in the water column. North Sea tides have a major influence on the turbulent dynamics of the system and are considered here. For better resolution of the modeled tides over the observation (TPXO9-atlas (1/30° resolution) [30]), we included the 2 most important harmonics, namely the solar (*S2*) and lunar semidiurnal tides (*M2*). Detailed GOTM specifications are presented in Table 1. The Baltic Sea tides are only small, and tidal turbulence is weak, and hence tides are neglected for the Baltic Sea setups.

*2.3. ECOSMO*

ECOSMO is a multicompartment complex biogeochemical numerical NPZD (nutrients, phytoplankton, zooplankton, and detritus) model, which resolves nutrient cycling, 2 groups of non-living organic matter (detritus (POM)) and dissolved organic matter (DOM)), and 5 biological functional groups consisting of 3 phytoplankton and 2 zooplankton state variables [22] (Figure 2). Particulate organic matter (POM) in marine environments is the sum of all kinds of labile, non-living particulate organic matter. Source terms for POM are the mortality components of phytoplankton and zooplankton ($MORT_{BIO}$), and they decrease as a function of temperature-dependent remineralization ($REMIN_{POM}$) and

feeding by zooplankton ($FEEDING_{z\text{-}POM}$) [22]. In ECOSMO, the concentration of POM is calculated as

$$\frac{d[POM]}{dt} = (1 - \alpha_{DOM})MORT_{BIO} - REMIN_{POM} - FEEDING_{z-POM} \qquad (4)$$

The mentioned above processes are taken into account for all species of plankton (microzoo-, mesozooplankton, and three phytoplankton classes (i.e., flagellates, diatoms and cyanobacteria)) (Figure 2) [22]. $\alpha_{DOM}$ is part of the mortality component of phytoplankton and zooplankton ($MORT_{BIO}$) turning to DOM.

DOM is the concentration of all kinds of labile-dissolved organic matter. The DOM pool grows due to phyto- and zooplankton mortality and declines as a function of remineralization [22]:

$$\frac{dDOM}{dt} = \alpha_{DOM}MORT_{BIO} - REMIN_{DOM} \qquad (5)$$

*2.4. Model Coupling, Initial and Boundary Conditions*

Hydrodynamic (GOTM), biogeochemical (ECOSMO) and chemical (POP model) modules are coupled through FABM. FABM allows to integrate biogeochemical processes seamlessly, independent of the hydrodynamic host model. Moreover, FABM facilitates the exchange of state variables between the host and any number of client models.

All of the implemented FABM processes (Figure 3) are classified as external or internal. External processes include exchange with atmosphere and sediment, and all related compartments (atmospheric PCB concentrations, ice concentration, and wind speed). The internal processes consider transformations of PCB in the water column, such as kinetic sorption of $PCB_{153}$ on OM (both DOM and POM) and degradation processes (photolytic and biological). In the end, all considered processes determine the amount of PCB in the dissolved phase ($PCB_{free}$). Thus, at the last stage, our model is able to estimate areas of high PCB concentration, where biota is potentially able to adsorb this type of pollutant.

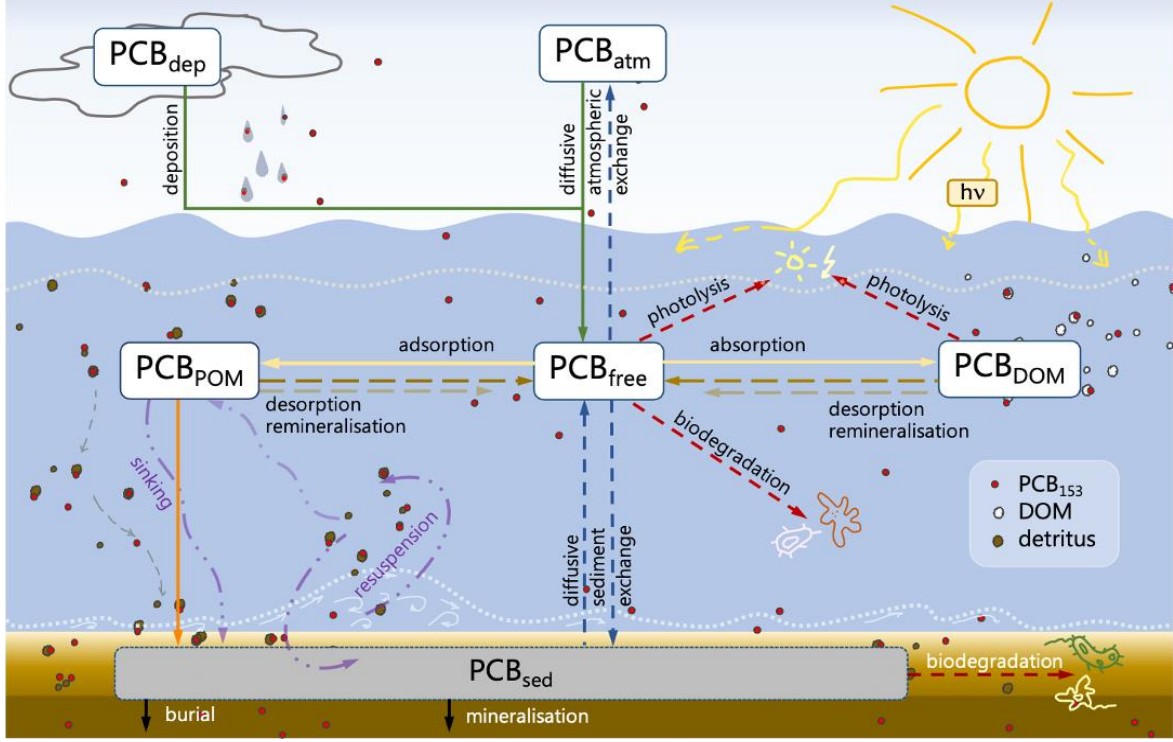

**Figure 3.** Main processes of $PCB_{153}$ transformation in current model. Arrows: green—input; yellow—phase transformation; purple—physical processes; red—removing of pollutants from the system.

Initial conditions of $PCB_{153}$ in the water column are prescribed as zero for all pollutants ($PCB_{free}$, $PCB_{DOM}$ and $PCB_{POM}$), and the system runs until it reaches equilibrium.

### 2.5. Chemical Model

#### 2.5.1. Particle Partitioning (Kinetic Sorption)

The general chemical structure of PCBs leads to a higher affinity to organic carbon matrices compared to water. In the presence of OM, they tend to accumulate on it rather than stay dissolved. This process is considered in the model via a simplified partitioning using only non-living organic matter (DOM and POM). Sorption on living matrices is neglected here; it has a complicated nature and requires several steps of biological accumulation, concentration and magnification, which is beyond the scope of the current research.

Sorption of PCB on OM is generally characterized by the organic carbon–water partitioning coefficient $K_{oc}$. This coefficient is determined by the chemical structure of both the pollutants and the absorbing matrix [5,10,11] and requires information about the direct chemical composition of each of the compartments involved in this process. That makes the determination of $K_{oc}$ a complicated task for modeling. However, the concept of linear free-energy relationships (LFERs) allows to estimate $K_{oc}$ based on the octanol–water coefficient $K_{ow}$. This coefficient is less complicated to measure, and the data of $K_{ow}$ for different pollutants are available in the scientific literature [4,10,11]. For different PCB congeners, $logK_{ow}$ varies from 4.46 ($CB_1$) to 8.18 ($PCB_{209}$) [11], which has an influence on their affinity for partitioning.

Here, we use the concept of sorption rates ($k_{sorp}$, $k_{desorp}$) instead of an equilibrium coefficient ($K_{oc}$). The concentration of pollutants bound to POM ($PCB_{POM}$) is computed based on the following equation (Equation (6)):

$$\frac{d[PCB_{POM}]}{dt} = \begin{aligned} &-(k_{desorp1} + k_{rem1})[PCB_{POM}] - (k_{bur} + k_{sed})[PCB_{POM}] \\ &+ k_{sorp1}[POM]\left[PCB_{free}\right] + k_{resusp}[PCB_{sed}] \end{aligned} \tag{6}$$

Parameters for kinetic sorption ($k_{sorp1}$) are calculated from $K_{oc}$ and $k_{desorp1}$ (as described by van Noort et al.; Jonker and Smedes) [31,32]. Rate constants for desorption ($k_{desorp1}$) are taken from [31] as slow desorption constants (Table 2).

**Table 2.** Rate constants describing process of $PCB_{153}$ sorption on organic matrices.

| | $k_{desorp} \cdot 10^{-6}$ (s$^{-1}$) | $k_{sorp}$ (s$^{-1}$) |
|---|---|---|
| DOM-$PCB_{153}$ | 3.8 [1] | 4.87 [3] |
| POM-$PCB_{153}$ | 5.22 [2] | 10.64 [3] |

Note: [1] Estimated. [2] P.C.M. van Noort et al./*Water Res.* **2003**, *37*, 2317–2322 [31]. [3] Calculated.

$k_{rem1}$ ($T$ °C) determines the process of detritus remineralization, which allows PCBs bound to POM to be released back into a dissolved form. This parameter is calculated in ECOSMO by the following Equation (7) [22]:

$$k_{rem1} = 2 \cdot 10^{-8}\left(1 + 20\left(\frac{T^2}{(13^2 + T^2)}\right)\right) \tag{7}$$

Another kind of OM which accumulates PCBs in this model system is DOM. Due to the complexity of measuring this phase, only few studies of PCBs on DOM [33] are available in the literature [4–6,34]. Particles of DOM have much smaller size than those of POM, and therefore a bigger reactive surface area [35]. That makes DOM a very good matrix for PCBs absorption.

In general, processes describing the sorption on POM are the same as those for DOM, but they employ different rates:

$$\frac{d[PCB_{DOM}]}{dt} = -k_{desorp2}[PCB_{DOM}] + k_{sorp2}[DOM]\Big[PCB_{free}\Big] - k_{rem2}[PCB_{DOM}] \quad (8)$$

The parameters $k_{sorp}$ and $k_{desorp}$ here were taken or calculated from the literature [4,6,10,32–35] as being DOM-related (Table 2).

$k_{rem2}$ reflecting DOM remineralization is calculated in ECOSMO as temperature ($T$ [°C]) dependent [22]:

$$k_{rem2} = 2 \cdot 10^{-7}\left(1 + 20\left(\frac{T^2}{(13^2 + T^2)}\right)\right) \quad (9)$$

2.5.2. Photolytic Degradation

The chemical structure of PCBs also determines possible photochemical transformations of these pollutants. In fact, PCB congeners with a higher extent of chlorination or a planar configuration of phenyl rings are more likely to directly absorb UV light [14,15]. $PCB_{153}$ has 6 chlorine atoms in the molecule and is affected by photolytic processes.

The photochemical transformation processes are very complicated, and the relevance of specific photolytic degradation pathways is still under discussion in the scientific community [35–38].

Under the process of direct photolysis, photons transfer energy directly to the PCB molecule. This process has a stepwise nature and shows a consequent dechlorination at every step [14]. Attacked by photons, over time, more chlorinated PCBs are turning to less and less chlorinated CBs. Wong and Wong [15] showed in their work that the pattern of photolytic dechlorination is the same, with a different quantum yield for every stage. Thus, decreasing a higher-chlorinated congener (e.g., $PCB_{153}$) will increase an amount of lower-chlorinated PCB (such as $PCB_{28}$). Wong and Wong's experiment was conducted in alcohol solvents; however, in an aqueous environment, PCBs can form hydroxybiphenyls (PCB-OH) [14]. In the current study, we only implement this process as a degradation of the initial PCBs, without considering the products of degradation.

Meanwhile, the photon flux can affect other particles in the water column. These particles receive some amount of energy and turn into an excited state. They become a new source of energy which they can transfer to the PCB molecule. The process is known as indirect photolysis [4]. This process is even more complex than direct photolysis. The implementation of it requires more research and data from the scientific community.

Here we implemented only direct photolysis of $PCB_{153}$, neglecting indirect photolysis. The approach implemented in the current research is taken as a mix from Schwarzenbach [4] and Vione [37]. Here, the kinetic rate of $PCB_{153}$ degradation is calculated as a parameter, related to the photon flux ($F_{PCB153}$) and quantum yield of the reaction ($Q_{153}$):

$$k_{photo} = Q_{153}\, F_{PCB153} \quad (10)$$

The photon flux absorbed by a pollutant is calculated from the irradiance at different wavelengths for the UV part of the spectrum (Equation (11)):

$$F_{PCB153} = \left(\lambda\, I\, 0.836 \cdot 10^{-2}\right)\left[\frac{\varepsilon_{153}(\lambda)\Big[PCB_{free}\Big]}{EXT_{TOT}}\right] \quad (11)$$

$F_{PCB153}$—photon flux [$\mu mol \cdot m^{-2} \cdot s^{-1}$ = $\mu E$];
$\lambda$—wavelength [m];
$I$—sunlight irradiance in UV range of spectrum [$W \cdot m^{-2}$];
$\varepsilon_{153}$—extinction coefficient for $PCB_{153}$ [$m^2 \cdot mol^{-1}$];

$EXT_{TOT}$—total extinction, includes marine water, DOM, detritus and phytoplankton [$m^2 \cdot mol^{-1}$].

Here the light penetration into the water column is calculated as a relevant fraction of available UV irradiation, decreasing with depth. This variable is calculated using 5 nm wavelength bins to allow for the implementation of wavelength-dependent attenuation coefficients.

Due to the lack of available data for extinction coefficients, in current simulations, $\varepsilon_{153}$ is a wavelength-independent parameter. However, this model is intended to be applicable for different hydrophobic POPs, and with the necessary data, photolysis can be implemented with wavelength-dependent extinction coefficients.

### 2.5.3. Biological Degradation

As was mentioned above, interactions between POPs and living organisms are extremely complex and depend on a wide range of parameters. Biological degradation is not an exception. This process is not only dependent on the structure of the pollutant but also strongly determined by the type of bacterial community and chemical composition of the environment, specifically by the oxygen content [16]. In the presence of oxygen, some microbial communities are able to break the structure of PCB molecules (oxidative degradation) [4,9,16]. This process is controlled not only by the class of aerobic bacteria, but also the stereochemical structure of the PCB molecule and its affinity to the exact enzyme type [16]. Boyle et al. showed in their study that for most aerobic bacteria, the oxidative ring cleavage mechanism (ORCM) takes place [17]. In this case, the decomposition of PCB by the bphA-enzyme is possible only if PCB has no substitutional chlorine in both ortho and meta positions on the same side of a ring (e.g., PCB$_{28}$) (Figure 4).

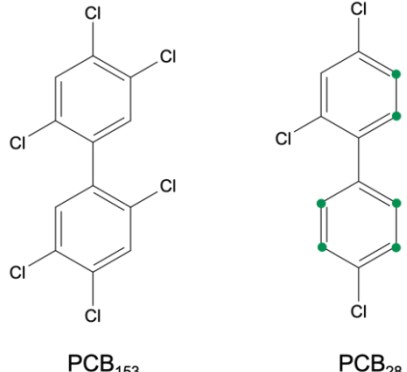

**Figure 4.** Availability of different PCBs chemical structure for aerobic biodegradation.

That means that aerobic bacterial degradation will have an impact on the concentration of such PCB congeners as PCB$_{28}$. However, chosen for current simulations, congener PCB$_{153}$ does not have a suitable structure for this process, and aerobic biodegradation is not applicable in this case.

At the same time, another process of biological transformation is available in the marine environment. This process can be rather called biological dechlorination than degradation. Without the presence of oxygen, other microorganism communities can also degrade PCBs, although by a completely different mechanism of reductive dechlorination [16]:

$$R\text{-}Cl + 2e^- + H^+ \rightarrow R\text{-}H + Cl^- \tag{12}$$

In this case, PCBs transform to a lower chlorinated pollutant, but stay in the environment. This path mainly occurs under anaerobic conditions, such as sediments or hypoxic waters [16].

The implementation of biological degradation into the model requires data of not only different bacterial communities and their concentration in the water column, but also

information about the bioavailability of PCBs. The potential mechanism of these processes is also only a hypothesis—kinetic and thermodynamic parameters determining this process are not available nowadays. At the same time, the biogeochemical model ECOSMO at the current stage does not resolve bacterial communities, and the model implementation requires simplification.

As a solution, we implemented a generalized process of biodegradation without specification of bacterial community, through parametrization from remineralization rates Equation (13):

$$k_{bio} = K_{BIO}\left(2 \cdot 10^{-8}\left(1 + 20\left(\frac{T^2}{(13^2 + T^2)}\right)\right)\left(10[DOM] + [POM]\right)\right) \tag{13}$$

where $T$ [°C] is the temperature and $K_{BIO}$ the parametrization coefficient.

Finally, a biological degradation rate in sediments is implemented as a constant and equal to $3.935 \cdot 10^{-10}$ s$^{-1}$ [39].

### 2.5.4. Freely Dissolved PCBs

All PCBs not bound to OM are freely dissolved in the water column (PCB$_{free}$) (Equation (14)). This phase of PCB is modulated by all relevant processes such as sorption–desorption and remineralization from non-living OM, and photolytical and biological degradation. Along with these processes, this fraction of PCB steadily increases by atmospheric influx (1st term) and diffusive exchange with the sediment (last 2 terms):

$$
\begin{aligned}
\frac{d[PCB_{free}]}{dt} = \quad & +\frac{F_d}{dz} - \left(k_{sorp1}[POM] + k_{sorp2}[DOM]\right)\left[PCB_{free}\right] \\
& + \left(k_{desorp1} + k_{rem1}\right)[PCB_{POM}] \\
& + \left(k_{desorp2} + k_{rem2}\right)[PCB_{DOM}] - (k_{photo} \\
& + k_{bio})\left[PCB_{free}\right] - k_{diff2}\left[PCB_{free}\right] + k_{diff1}[PCB_{sed}]
\end{aligned}
\tag{14}
$$

$\frac{F_d}{dz}$—atmospheric flux of pollutants (detailed description is below (Section 2.6.1));
$k_{diff(1,2)}$—rates of PCB$_{153}$ diffusive exchange with dissolved in water pollutants (detailed description is below (Section 2.6.2)).

The fraction of PCB$_{free}$ is relevant because it is the most susceptible to degradation processes (e.g., photolysis and biodegradation) and also to bioaccumulation.

### 2.6. Boundary Forcing and External Processes

### 2.6.1. Air–Sea Exchange

As the model lacks horizontal transport, the exchange of PCBs with the atmosphere is the only source of these pollutants for the water column. The current version of our model considers not only diffusive exchange with the atmosphere, but also the deposition of particulate PCB$_{153}$. Deposition is considered as a process of inflow of PCB on dissoluble particles (e.g., elemental carbon and organic aerosols). Once these pollutants reach surface waters, they contribute to the pool of PCB$_{free}$.

Atmospheric concentration and deposition fields were taken from the EMEP database [26].

The amount of PCBs, which come from the atmosphere, depends on several factors. Not only the direct atmospheric concentrations and deposition are important, as the ice coverage, wind speed at the surface, temperature and the dissolved PCB$_{free}$ fraction also influence the process.

The net direction of the air–sea flux for such hydrophobic substances as PCBs depends on the concentration gradient between the liquid and gaseous phases. To calculate the net diffusive flux ($F_d$) of a pollutant between air and water, the mixed approach from Friedmann and Selin [40] and Odabasi [41] was chosen. According to these authors, the

net flux can be calculated based on the Henry constant, mass transfer coefficient and both concentrations in air and in water (Equation (15)):

$$\frac{d[C_{atm}]}{dt} = F_d = \frac{\left(\frac{C_a}{H} - [PCB_{free}]\right)}{\frac{1}{k_w} + \frac{1}{(Hk_a)}} = \frac{\left(C_a - [PCB_{free}]H\right)}{\frac{1}{(k_wH)} + \frac{1}{k_a}} \tag{15}$$

$k_a$—mass transfer coefficient in air [m·s$^{-1}$];
$K_w$—mass transfer coefficient in water [m·s$^{-1}$];
$C_a$—PCB concentration in air [pg·m$^{-3}$];
$PCB_{free}$—PCB concentration in water [pg·m$^{-3}$].

Here $H$ is a temperature-dependent constant for PCB$_{153}$ according to Henry's law. This parameter is implemented into the model by Equation (16) (Table 3):

$$H = exp\left(\frac{-\Delta H}{RT} + \frac{\Delta S}{R}\right) \tag{16}$$

$\Delta H$—enthalpy of dissolution [J·mol$^{-1}$];
$\Delta S$—entropy of dissolution [J·mol$^{-1}$·K$^{-1}$];
$R$—gas constant [J·K$^{-1}$·mol$^{-1}$];
$T$—temperature [K].

**Table 3.** Parameters, used in calculations of air–water exchange.

| Parameter (Units) | Value | Source |
|---|---|---|
| MW$_{air}$ (g·mol$^{-1}$) | 28.97 | Calculated |
| MW$_{PCB}$ (g·mol$^{-1}$) | 360.88 | Calculated |
| MV$_{air}$ (cm$^3$·mol$^{-1}$) | 20.1 | Calculated |
| MV$_{PCB}$ (cm$^3$·mol$^{-1}$) | 0.080276 | Calculated |
| $\nu_{H2O}$ (cm$^2$·s$^{-1}$) | 0.0089 | Schwarzenbach [4] |
| D$_{aH2O}$ (cm$^2$·s$^{-1}$) | 0.3 | Schwarzenbach [4] |
| D$_{wCO2}$ (cm$^2$·s$^{-1}$) | 0.00002 | Schwarzenbach [4] |
| $\Delta$H (PCB$_{153}$) (J·mol$^{-1}$) | 66,100 | Bamford 2000 [18] |
| $\Delta$S (PCB$_{153}$) (J·mol$^{-1}$·K$^{-1}$) | 190 | Bamford 2000 [18] |

Following the Odabasi concept, these parameters were calculated based on wind speed and specific diffusivity. Wind speed data for current simulations were taken from the ECMWF ERA5 dataset (0.25°/hourly resolution):

$$k_w = \left[\frac{(0.24u_{10}^2 + 0.061u_{10})}{3600}\right]\left[\frac{D_{wi}}{D_{wCO_2}}\right] \tag{17}$$

$$k_a = [0.2u_{10} + 0.3]\left[\frac{D_{ai}}{D_{aH_2O}}\right] \tag{18}$$

$u_{10}$—wind speed at 10 m above a water [m·s$^{-1}$];
$D_{wi}$ and $D_{ai}$—diffusivities of PCB in the air and water [cm$^2$·s$^{-1}$];
$D_{wCO2}$—CO$_2$ diffusivities in the water [cm$^2$·s$^{-1}$];
$D_{aH2O}$—H$_2$O diffusivities in the air [cm$^2$·s$^{-1}$].

These parameters were taken from Schwarzenbach [4] as constants (Table 3), while $D_{wi}$ and $D_{ai}$ were calculated by the following equations [41]:

$$D_{ai} = \frac{T \cdot 10^{-3}\left[\frac{1}{MW_{air}} + \frac{1}{(MW_{PCB} \cdot 10^3)}\right]^{0.5}}{P\left(V_{air}^{0.33} + V_{PCB}^{0.33}\right)^2} \tag{19}$$

$$D_{wi} = \frac{13.26 \cdot 10^{-5}}{P \left(100 \nu_{H_2O}\right)^{1.14} V_{PCB}^{0.589}} \tag{20}$$

$MW_{air}$, $MW_{PCB}$—molar weight of an air and chosen PCB [g·mol$^{-1}$];
$V_{air}$, $V_{PCB}$—molar volume of an air and chosen PCB [cm$^3$·mol$^{-1}$];
$P$—air pressure [atm];
$\nu_{H2O}$—kinematic viscosity (at 25 °C) [cm$^2$·s$^{-1}$].

The presence of surface water ice affects not only the availability of incoming atmospheric pollutants, but also the general patterns of PCB distribution. Current simulations include ice as the fraction of ice coverage. This parameter is implemented from ECOSMO simulations [22]. ECOSMO provides a partial sea ice cover exchange scaling by $(1 - A_i)$, where $A_i$ is the sea ice compactness (the part of the cell covered by sea ice). The atmospheric flux ($F$) is also scaled by the ice cover, according to Equation (21). Here, $F_{ow}$ is the open water flux:

$$F = (1 - A_i) \cdot F_{ow} \tag{21}$$

2.6.2. Sedimentation, Resuspension and Burial

The process of sedimentation is driven by sinking particles. The sinking process of PCB$_{153}$ on particles is included in our simulations by a constant sinking velocity of 1 m d$^{-1}$ [22]. PCB$_{DOM}$ and freely dissolved pollutants are not sinking. Therefore, sinking velocities are implemented only for PCB on POM.

In our model, several processes determine the amount of PCBs stored in sediments: direct sedimentation, diffusive exchange with the water column, degradation and burying.

The concentration of PCB$_{153}$ in the sediment is presented in the model as

$$\begin{aligned}
\frac{d[PCB_{sed}]}{dt} = \ & -(k_{bur} + k_{deg} + k_{rem3})[PCB_{sed}] - k_{diff1}[PCB_{sed}] \\
& - k_{resusp}[PCB_{sed}] + k_{diff2}\left[PCB_{free}\right] + k_{sed}[PCB_{POM}]
\end{aligned} \tag{22}$$

$k_{bur}$—burial rate of PCB$_{153}$ (equal for a POM burial rate in ECOSMO);
$k_{deg}$—rate constant of pollutant degradation in sediments;
$k_{diff(1,2)}$—rates of PCB$_{153}$ diffusive exchange with dissolved in water pollutants;
$k_{resusp}$—resuspension rate (bottom shear-stress-dependent);
$k_{sed}$—sedimentation rate of PCB$_{POM}$;
$PCB_{POM}$—concentration of PCB$_{153}$ on POM [pg·m$^{-3}$];
$PCB_{free}$—dissolved in water PCB$_{153}$ [pg·m$^{-3}$];
$k_{rem3}$—rate of sediment remineralization.

The rate of sediment remineralization is calculated as

$$k_{rem3} = 2 \cdot 10^{-8} \, exp(temp_C \cdot T) \tag{23}$$

$temp_c$—temperature control factor [°C$^{-1}$];
$T$—temperature [°C].

All rate constants used in Equation (22) are listed in Table 4.

**Table 4.** Rate constants, used in modeling PCB$_{153}$ sedimentation.

|  | $k_{bur} \cdot 10^{-9}$ (s$^{-1}$) | $k_{diff} \cdot 10^{-10}$ (s$^{-1}$) | $k_{deg} \cdot 10^{-10}$ (s$^{-1}$) | $k_{ex} \cdot 10^{-5}$ (s$^{-1}$) * |
|---|---|---|---|---|
| Sediment PCB$_{153}$ | 1.157 [1] | (1) 0.139 [2] | 3.935 [2] | 28.935 [1] |
| PCB$_{free}$ | – | (2) 4.05 [2] | – | – |
| PCB$_{POM}$ | – | – | – | 4.051 [1] |

Note: * Exchange rate constants: $k_{resusp}$ for PCB$_{sed}$ and $k_{sed}$ for PCB$_{POM}$. [1] ECOSMO [22]. [2] Jay A. Davis 2009 SETAC [39].

Finally, resuspension is triggered when the friction velocity near ground surpasses a critical velocity (Equation (24)). The friction velocity ($u_*^b$) is simulated in GOTM, while critical velocity ($u_{cr}^b$) is set as 0.07 m s$^{-1}$ (Equation (25)):

$$u_*^b > u_{cr}^b \tag{24}$$

$$u_*^b = r\sqrt{U_1^2 + V_1^2} \tag{25}$$

$U_1$, $V_1$—components of mean velocity at the center of the lowest cell.

### 2.7. Regional Characteristics for Model Implementation

To investigate the influence of hydrodynamic and biogeochemical characteristics on the transformation and mixing processes of PCBs in the water column, the column model was implemented for 4 hydrodynamically different locations resembling conditions from different locations in the North and Baltic Seas using the framework of the hydrodynamic column model GOTM [29]. To illustrate the influence of hydrodynamics on the fate of PCBs, we initialized and forced the water column model with a realistic water depth, initial tidal conditions and atmospheric forcing for regions in the North and Baltic Seas (Figure 5). Model specifications and parametrizations of the chosen regions are presented in Section 2.2.

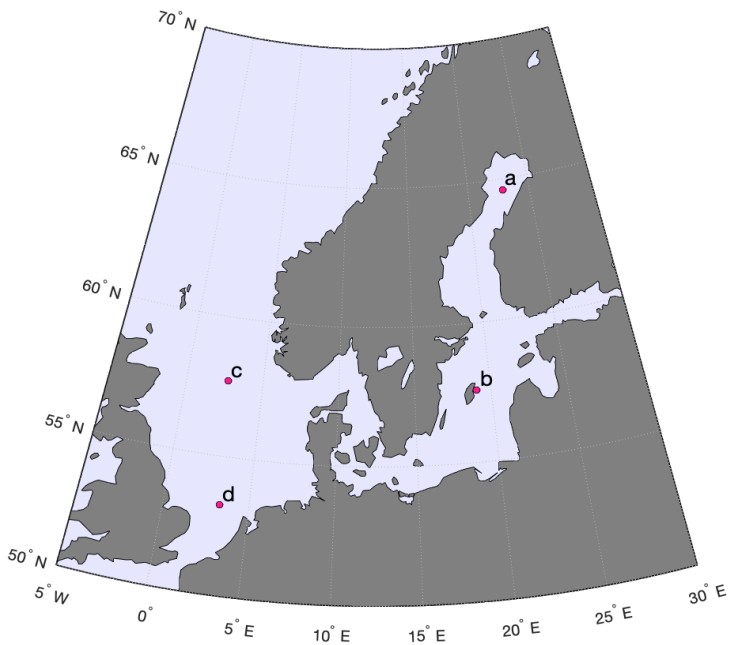

**Figure 5.** The map of chosen locations for a model runs: (**a**,**b**)– the Baltic Sea; (**c**,**d**)—the North Sea.

These regions were chosen as a study region due to the high quantity of chosen pollutants in the system [25]; large differences in hydrodynamic, meteorological and biological conditions [42–44]; and also, a high risk for bioaccumulation due to the high primary production, negatively affecting food safety [12].

The Baltic Sea is a semi-closed sea, which has relatively low salinity. This feature is determined by the high river runoff and limited connection with the open ocean (only through North Sea) [45]. That means that the amount of PCBs introduced from land over the last century was high, while the exchange with the open ocean was only minimal. For the reason that PCBs have long residence time, they stay accumulated in this marine system.

In the Baltic Sea, two locations were selected for model simulations. Firstly, the Central Baltic (Figure 5b) is a brackish water region which is characterized by a stable year-

round 2-layer stratified system, with fresh surface water and a more saline, often anoxic, lower layer [46,47]. The upper layer will experience winter and summer thermocline development and occasionally sea ice development in severe winters. The lower layer is practically isolated from seasonal variations. The second characteristic environment studied is chosen to be similar to the conditions in the northernmost part of the Baltic, the Bothnian Bay (Figure 5a). This area is characterized by almost fresh water conditions and regular seasonal ice cover over several months [47].

The North Sea is characterized by tidal activity and subdivided into different areas defined by mixing processes [42,48]. We chose two regions, with widely different conditions.

The northern region is located in the central-northern part (Figure 5c), in the area determined by seasonal stratification [42]. This region is 110 m deep and has tides with *M2* amplitude 0.41 m and *S2*—0.16 m [30,49]. At the same time, this region is influenced by the North Atlantic inflow, and has lower nutrient content and biological production than the southern part [22].

The southern part (Figure 5d) is a shallow area (41 m), strongly affected by tides [42]. Here, their *M2* amplitude can reach up to 0.7 m, and *S2*—0.25 m [30]. These 2 characteristics cause high turbulence leading to constant mixing and resuspension. The water column stays mixed over the entire year [42]. Due to conditions of warm mixed water and abundance of nutrients, in this region, the primary production is high [48].

## 3. Results and Discussion

Here, we investigate the processes governing PCB distribution patterns in the four aforementioned regions. There are two major dynamics determining the fate of PCB in the coastal ocean.

The first one is biological production which leads to high OM concentrations transforming PCB from the dissolved to the absorbed phase. Moreover, POM scavenges PCB from the surface layer due to gravitational settling. The associated change in the atmosphere ocean gradient of PCB concentrations at the interface leads to an increased atmospheric influx via diffusive exchange. Since the atmosphere is the main source of PCB in the 1D model, the amount of OM at the surface plays an important role in the total amount of PCB in the system. Resuspension can also increase the amount of suspended OM, and thus enhance the process of pollutant accumulation directly on it by increasing nutrient concentrations, leading to higher primary production in nutrient limited regions.

The second driving processes are turbulence regime and stratification. Depending on the mixing condition, sediments can become a permanent or seasonal sink for PCB. Independent of the long-term fate, stratification leads to the removal of PCB from the surface layer due to particle settling and will thus also impact the air–sea exchange. In regions with seasonal stratification, resuspension events can lead to pronounced peaks in PCB concentrations and can therefore have a major impact on bioaccumulation.

Here we evaluated four regions with different conditions. The two model locations in the Baltic Sea represent a stratified region with a deep basin (Gotland Deep) and a mixed shallower region in the north, which has seasonal ice coverage (Bothnian Bay). The Gotland basin area is the deepest region of the Baltic Sea (250 m), which has a stable stratification [44]. Surface and bottom mixed layers are permanently separated by a halocline. Thus, resuspension events are rare and do not affect the stratified surface layer. In the Bothnian Bay location, winter sea ice formation restricts the amount of atmospheric PCB introduced from the atmosphere.

In contrast to the Baltic Sea, the North Sea is a much shallower shelf sea with no deep basins. More importantly, it is connected to the Atlantic Ocean and physical and biological processes determine whether PCBs originating from the continent are deposited into the sediments or transported into the open ocean. For our analysis, we model PCB cycling at two locations in the North Sea that differ due to two processes: namely, stratification/mixing and primary production/biological pump. The NNS is seasonally stratified

with high primary production. The SNS is mixed throughout the year and exhibits even higher primary production. The stronger vertical mixing is caused by the shallow depth and higher tidal amplitudes in the SNS. When the water column is mixed, the bottom turbulence (sheer stress) prevents PCB from accumulating in the sediments. In the case of seasonal mixing, this can lead to pronounced resuspension peaks. Finally, the high biological production leads to high concentrations of dissolved and particulate organic matter. This strongly influences the speciation of PCB and causes sedimentation through PCB bound to detritus (detritus vector). For the Baltic Sea, speciation influences the air–sea exchange, as only the dissolved form exchanges between atmosphere and ocean.

For the North Sea, we evaluate two model scenarios for each region. Besides the default scenario, we also ran the model without tidal forcing to show the impact the tides have on the fate of PCB in this region. The turbulence introduced by tides is key for determining whether sediments are a sink for PCB in this region and have implications for their long-range transport. On the one hand, the tides lead to more resuspension, reducing net sedimentation. On the other hand, the increased availability of nutrients in a largely N-limited environment leads to increased primary production which in turn leads to more POC and thus increases the sedimentation flux.

### 3.1. The Deep Region with Permanent Stratification—The Gotland Deep

Primary production in this area begins in May and produces high concentrations of POM and DOM from decaying phytoplankton. These adsorb $PCB_{free}$ and due to the settling of $PCB_{POM}$, effectively scavenge PCBs from the surface layer toward the ocean floor (Figure 6c). Due to low turbulence and weak currents, there is rarely any resuspension at the bottom of Gotland Basin, and PCBs accumulate permanently in the sediment (Figure 6e).

In this region, subsurface primary production becomes nutrient-limited early in summer. As a result, a clear seasonality is seen in the $PCB_{free}$ distribution—in early summer, detritus removes pollutants from the dissolved form with the beginning of biological production (Figure 6c,f). Because cyanobacteria bloom in late summer and autumn, this biological pump is active almost until the end of the year, albeit at a lower level (Figures 6c and 7a). Thus, the stratified surface layer becomes 'cleaned up' from pollutants, and due to the more pronounced concentration gradient, the atmospheric PCBs flux increases (Figure 6f,d). PCBs entering from the atmosphere have no matrix to attach to. Thus, exchange with the atmosphere slowly saturates the upper layer with PCBs over time (Figure 6f). Stratification, lack of detritus and high input of atmospheric PCBs create a surface layer of $PCB_{free}$ building up until the next biological production cycle. Finally, the high $PCB_{free}$ concentration at the onset of biological production is a hotspot of bioavailability (Figure 6b,f).

### 3.2. Permanently Mixed Region with Low Atmospheric Input and the Late Onset of Production Due to Sea Ice Cover—The Bothnian Bay

This region exhibits seasonal sea surface ice formation during wintertime, and primary production starts even later in the year. However, in this northern location in the Baltic Sea, the water column is well-mixed (Figure 7a).

This region exhibits seasonal sea surface ice formation during wintertime, and primary production starts even later in the year. Being more remote in the northern part of the Baltic Sea, this region is characterized by lower atmospheric PCB concentrations, and during the period of ice coverage, the atmospheric PCB flux drops down to almost zero. (Figures 6d and 7d).

In this region, the water column is mixed throughout the entire year due to convective mixing (Figure 7a). In winter, before ice development begins, newly incoming atmospheric PCB directly sorbs on OM and then mixes in the water column (Figure 7b,c). A small fraction of these pollutants come back into the dissolved form during the remineralization of OM (Figure 7f). However, at the end of summer, a thin stratified layer directly at the surface is not affected by this process for a short period of time (Figure 7a,f). The Arctic conditions leave only a short window for biological production and the associated OM

production. These physical and biogeochemical conditions allow atmospheric PCB to dissolve into the water and stay in PCB$_{free}$ form. In our model, we did not include the PCB interactions with ice, but it can be assumed that a fraction of atmospheric PCBs get stored in the sea ice, potentially leading to an even more pronounced spring peak in PCB concentrations when the ice melts.

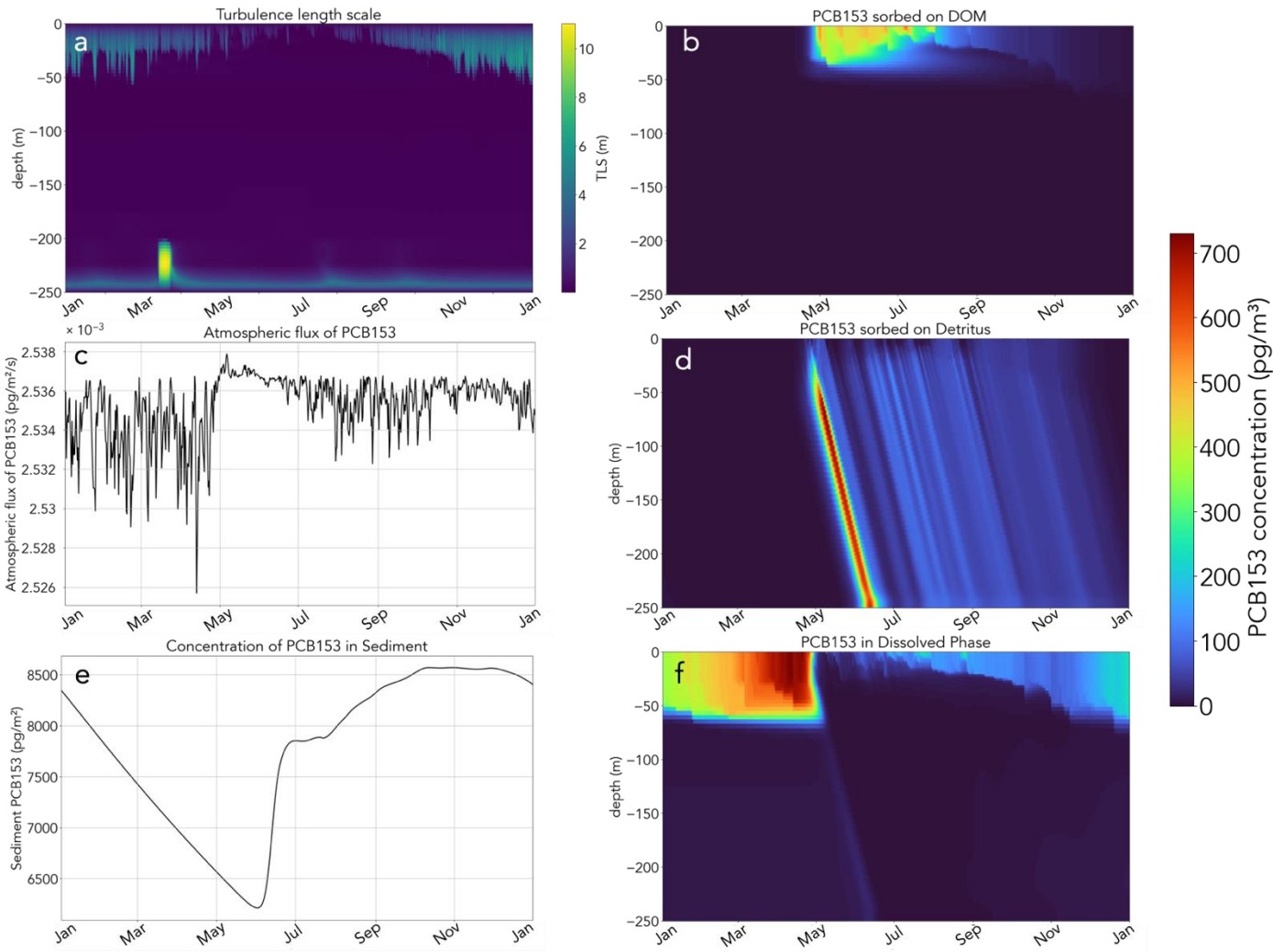

**Figure 6.** Results of simulation in the Gotland Basin (year 2001): (**a**)—TLS (m); (**b**)—PCB$_{DOM}$ concentrations (pg m$^{-3}$); (**c**)—atmospheric flux of PCB$_{153}$ (pg m$^{-2}$ s$^{-1}$) (direction of flux—from atmosphere to water); (**d**)—PCB$_{POM}$ concentrations (pg m$^{-3}$); (**e**)—sediment concentration of PCB (pg m$^{-2}$); (**f**)—PCB$_{free}$ concentration (pg m$^{-3}$).

### *3.3. The North Sea—Tides Influenced Area with High Primary Production*
Seasonally Stratified—Northern North Sea (NNS)

The key process governing the differences between model runs with and without tides is the additional energy introduced by the tides which increases the turbulence and thus vertical mixing. Figure 8 depicts the seasonal turbulence for the model run with and without tides.

The tides raise the depth and turbidity of the bottom mixed layer, leading to a breakdown of the stratification during winter (Figure 8b). This also increases sediment resuspension, introducing more particulate OM and nutrients into the water column. In both model runs, the upper and bottom mixed layers are separated during summer.

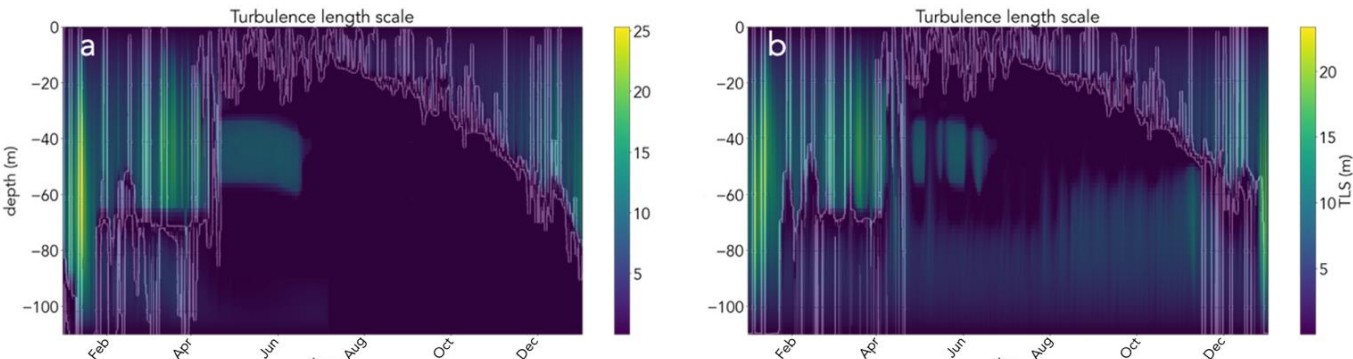

**Figure 7.** Results of simulation in the Bothnian Bay (year 2001): (**a**) TLS; (**b**) PCB$_{DOM}$ concentrations (pg m$^{-3}$); (**c**) atmospheric flux of PCB$_{153}$ (pg m$^{-2}$ s$^{-1}$) (direction of flux—from atmosphere to water); (**d**) PCB$_{POM}$ concentrations (pg m$^{-3}$); (**e**) sediment concentration of PCB (pg m$^{-2}$); (**f**) PCB$_{free}$ concentration (pg m$^{-3}$).

**Figure 8.** TLS (m) and surface mixed layer (SML) depth in 2 scenarios: (**a**) without tides, (**b**) with tides (the water column stays stratified during summer season).

　　PCB cycling varies strongly between the two model runs. Common patterns are, like in the Baltic Sea, the scavenging of PCB from the surface layer due to settling of POM. In

the run without tides, the PCB accumulates in the sediment and is partially resuspended during a short time of mixing in January (Figure 9(a2,a3)).

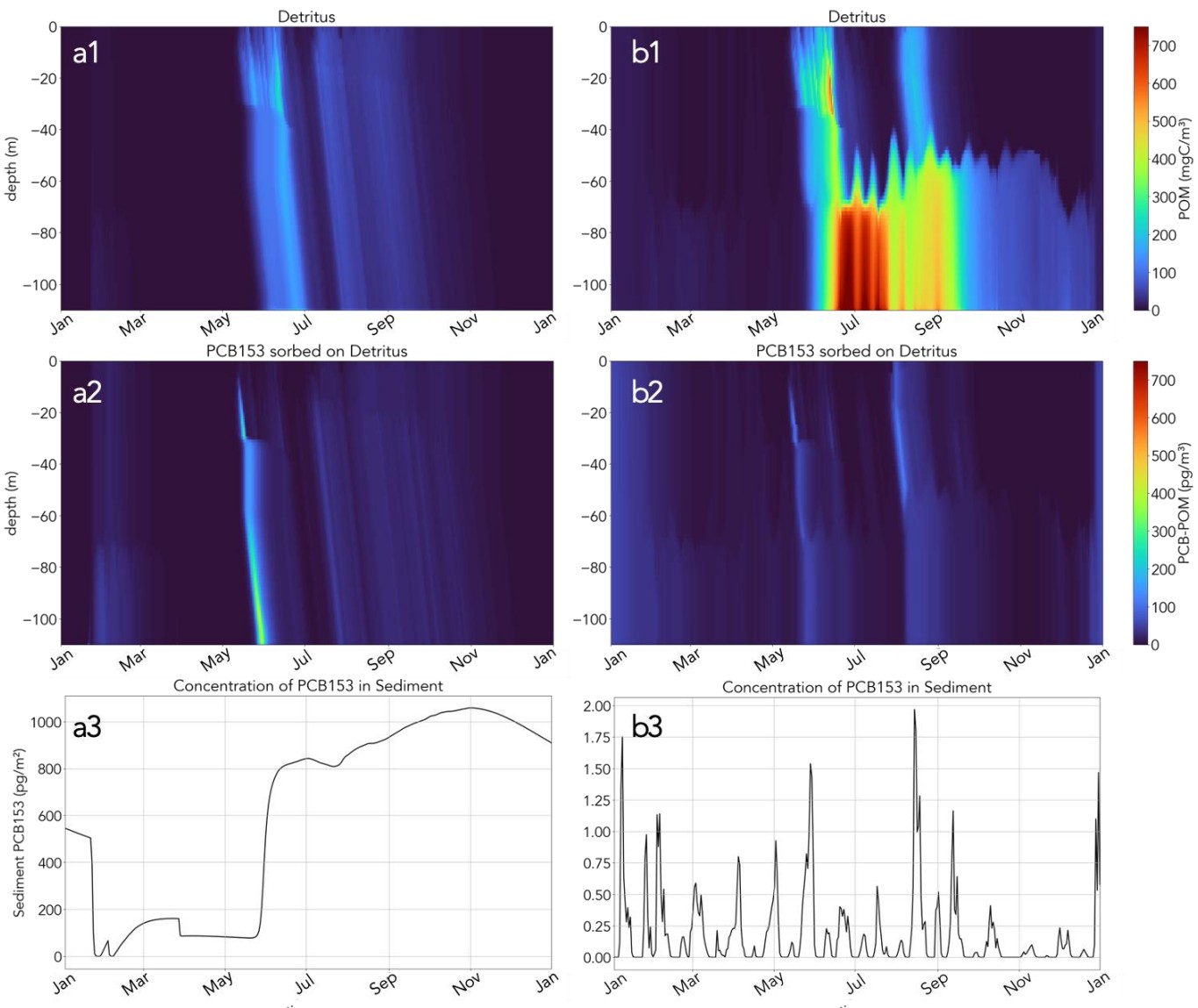

**Figure 9.** Results of simulations for: (**a1**,**a2**,**a3**) no-tide scenario, (**b1**,**b2**,**b3**) tide scenario (NNS, 2001 year). Concentrations of (**a1**,**b1**) detritus (mgC m$^{-3}$); (**a2**,**b2**) PCB$_{POM}$ (pg m$^{-3}$); and (**a3**,**b3**) PCB in sediments (pg m$^{-2}$).

In the case with tides, there is enough energy in the system to resuspend PCBs from the sediment, which gets mixed below the stratification (Figure 9(b2,b3)). The result is little or no annual net accumulation of PCBs in sediments (Figure 9(b3)) and concentrations of suspended PCB$_{POM}$ are generally higher in the bottom layer (Figure 9(a2,b2)). In this scenario, the presence of detritus during winter increases scavenging, and dissolved PCB$_{free}$ becomes, at times, completely depleted. This leads to an increased PCB flux from the atmosphere (Figure 10(b2)). The atmospheric PCB is quickly absorbed to the abundant OM, which in turn increases the influx of atmospheric PCB further.

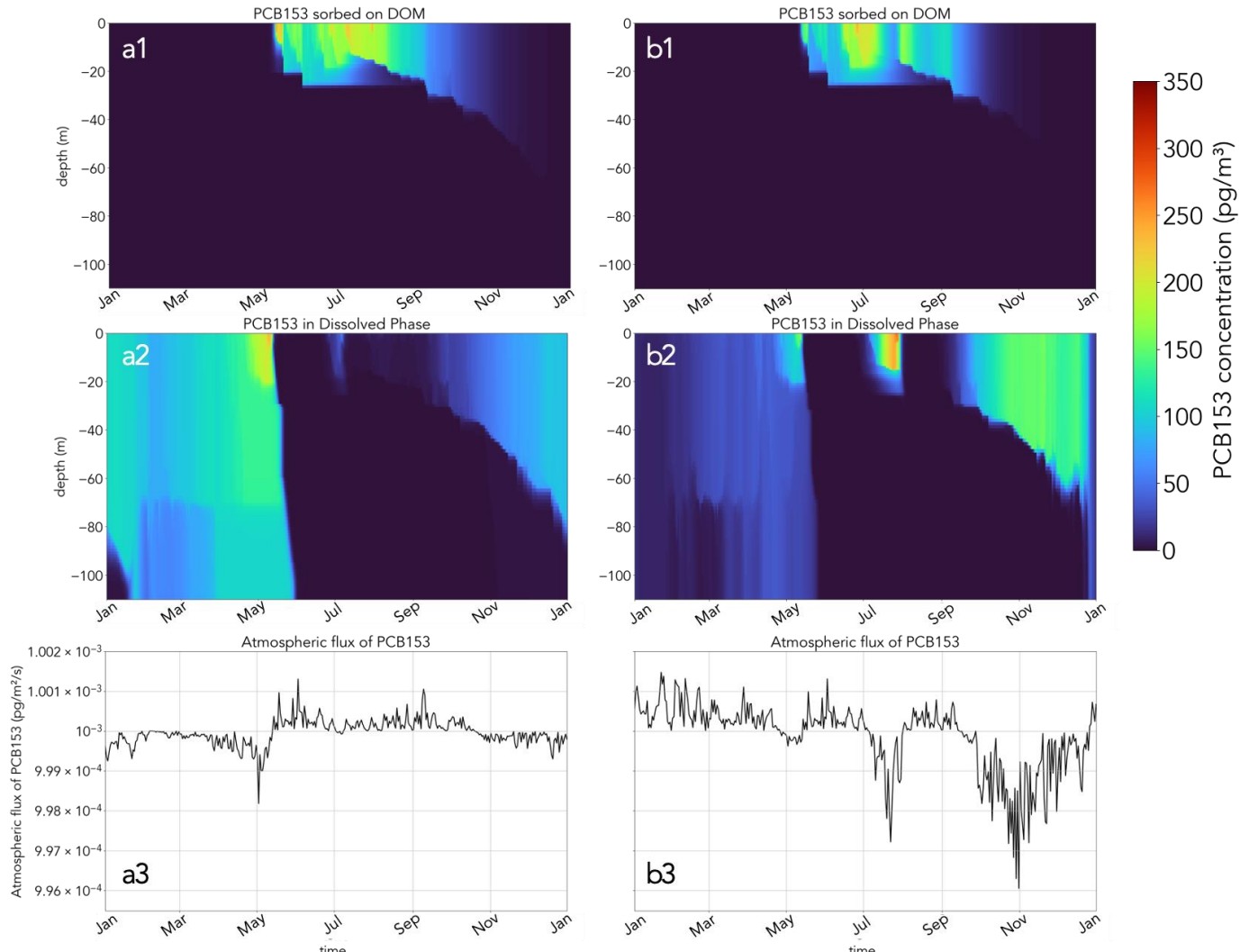

**Figure 10.** Results of simulations for (**a1,a2,a3**) no-tide scenario, (**b1,b2,b3**) tide scenario (NNS, 2001 year). Concentrations of (**a1,b1**) $PCB_{DOM}$ (pg m$^{-3}$); (**a2,b2**) freely dissolved $PCB_{153}$ (pg m$^{-3}$); (**a3,b3**) atmospheric flux of $PCB_{153}$ (pg m$^{-2}$ s$^{-1}$) (direction of flux—from atmosphere to water).

In the no-tide scenario, winter wind-driven short-term resuspension events leads to very high dissolved PCB concentrations during spring (before phytoplankton bloom) (Figure 10(a2)). Due to the absence of OM and well-mixed conditions, entering the water column atmospheric POPs contribute to $PCB_{free}$ fraction and distribute along with the depth.

Tidal activity does not affect the surface mixing layer during the summer season; near surface $PCB_{DOM}$ exhibits similar patterns of distribution and concentration in both scenarios (Figure 10(a1,b1)).

The PCB concentration difference between the two runs is depicted in Figure 11. In the tidal scenario, resuspension prevents pollutants from being stored in sediment (Figure 9). The bottom layer stays mixed throughout the year, and mixing does not release additional pollutants from the sediment during winter. As a result, the tidal activity decreases the concentration of freely dissolved PCB in the NNS area during the winter season (Figure 11).

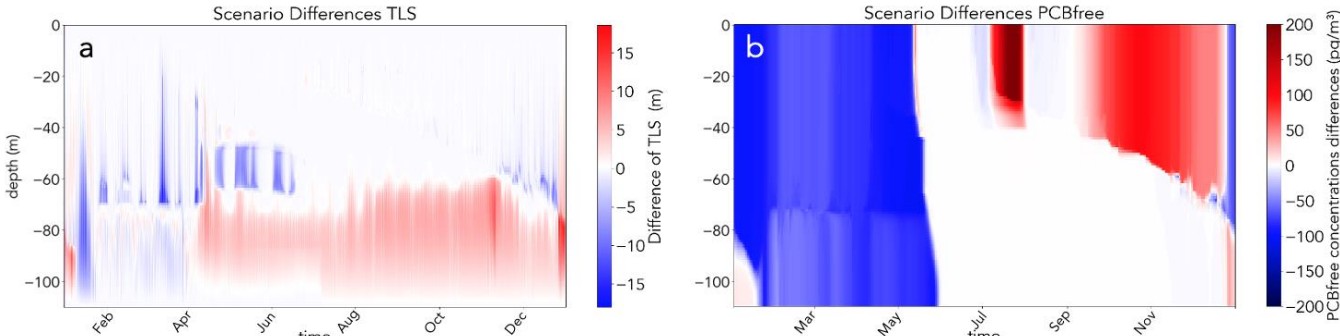

**Figure 11.** Differences between tide–no-tide scenarios for NNS region: (**a**) in turbulence length scale; (**b**) in PCB$_{free}$ concentration.

### 3.4. Permanently Mixed—Southern North Sea (SNS)

In the Southern North Sea (SNS), the tidal range is higher the in NNS and the water column is shallower (41.5 m). Under these conditions, the bottom mixed layer reaches the surface mixed layer even during summer (Figure 12b), and the entire water column is mixed from surface to bottom. Without tides, the water column becomes stratified during summer (Figure 12a).

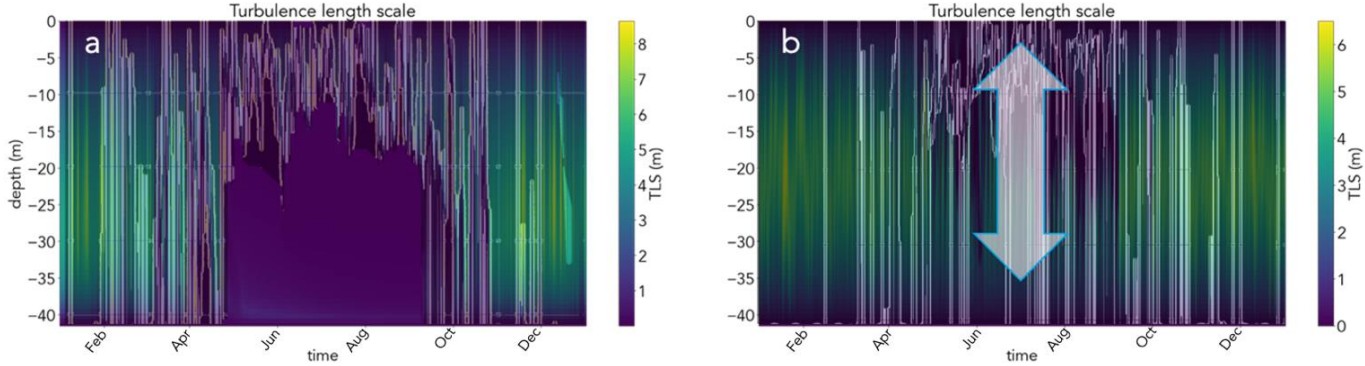

**Figure 12.** TLS (m) and SML depth in 2 scenarios: (**a**)—without tides, (**b**)—with tides (SML and BML (bottom mixed layer) meet during summer season—water column is permanently mixed).

This produces a situation where resuspended OM from the sediment is mixed all the way to the surface. Thus, there is an abundance of OM over the entire year that decreases dissolved PCB$_{free}$ concentrations and increases the atmospheric influx. This means that the tides effectively change the equilibrium between the atmosphere and ocean, and there is more PCB in the water when the tides are considered. Moreover, due to the constant mixing, tides prevent any significant sedimentation (Figure 13(b3)).

There is a pronounced seasonality of PCB$_{free}$ concentrations with a similar pattern in both cases (Figure 14(a3,b3)). In the tidal scenario, nutrients resuspended from sediment enhance biological production at the subsurface layer, therefore increasing the content of OM in the water column. When productivity stops at the end of autumn, and remaining OM remineralizes, releasing absorbed pollutants PCB$_{POM}$ back to the dissolved phase. Thus, PCB$_{free}$ concentrations, which are almost depleted at this point, begin to build up again. PCB$_{free}$ peaks the next year when fresh OM is produced again. This peak is more pronounced in the tidal scenario due to there being more PCB on OM.

In summary, the tides increase the total PCB load and especially the bioavailable PCB$_{free}$ concentration due to the additional OM resulting from sediment resuspension (either directly or through additional nutrients driving primary production). The high content of OM, mixed in the water column throughout the year, keeps pollutants in the ocean. PCBs absorbed on OM release to a dissolved form, increasing PCB$_{free}$ concentrations

over time. This is the opposite of what we found for $PCB_{free}$ in the NSN. There, the tides prevent sedimentation and thus there is no spring peak through resuspension (Figure 14).

High content of OM due to increased biological production, accumulates PCB from its dissolved form, while higher turbidity in the water column prevents these pollutants from being stored in the sediment. Both DOM and detritus slowly release PCBs back to the dissolved phase during remineralization. As a result of all these processes in the SNS, the concentration of $PCB_{free}$ increases in the tidal scenario, compared with the no-tidal scenario (Figure 15).

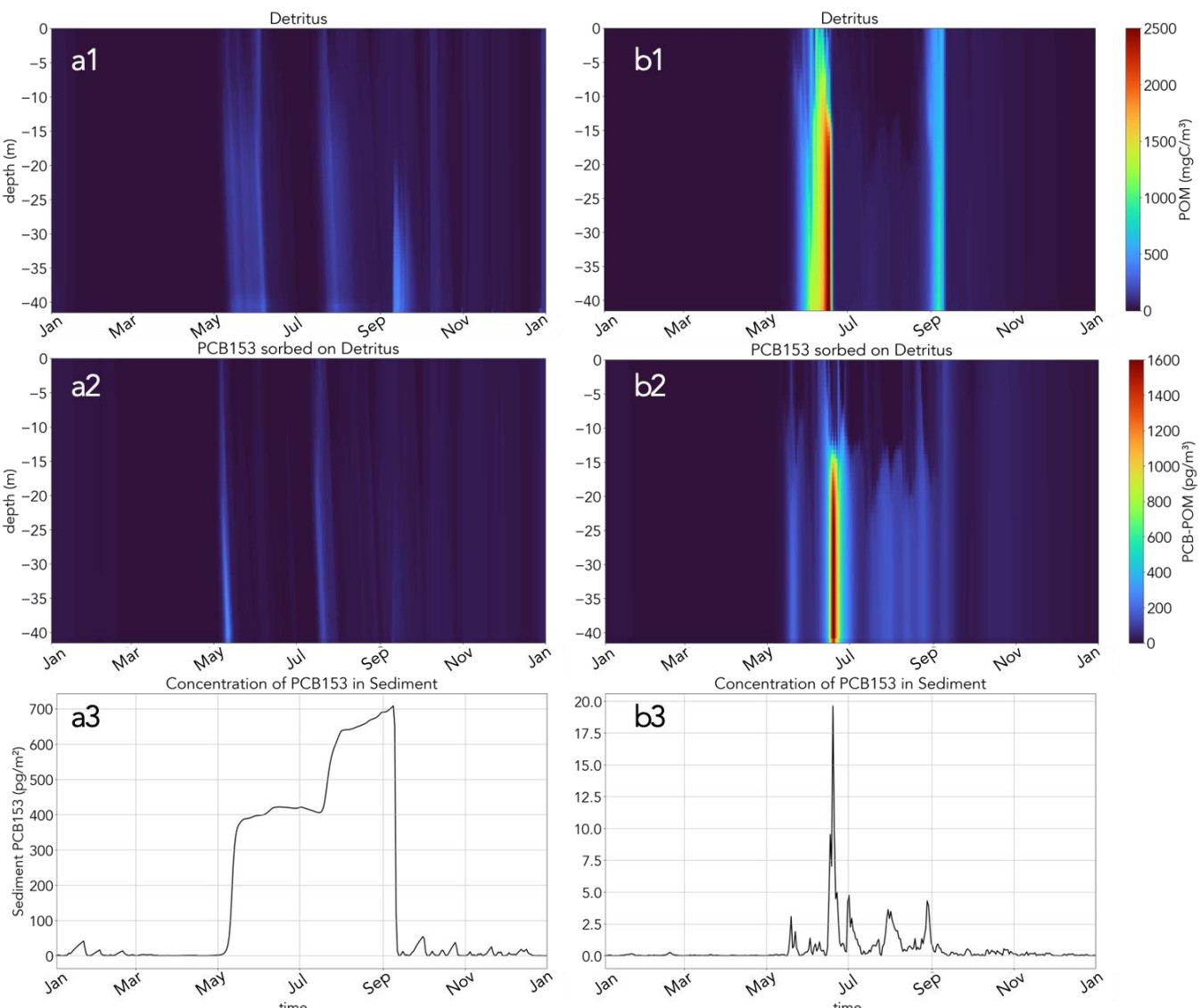

**Figure 13.** Results of simulations for (**a1**,**a2**,**a3**) no-tide scenario, (**b1**,**b2**,**b3**) tide scenario (SNS, 2001 year). Concentrations of (**a1**,**b1**) detritus (mgC m$^{-3}$); (**a2**,**b2**) $PCB_{POM}$ (pg m$^{-3}$); (**a3**,**b3**) PCB in sediments (pg m$^{-2}$).

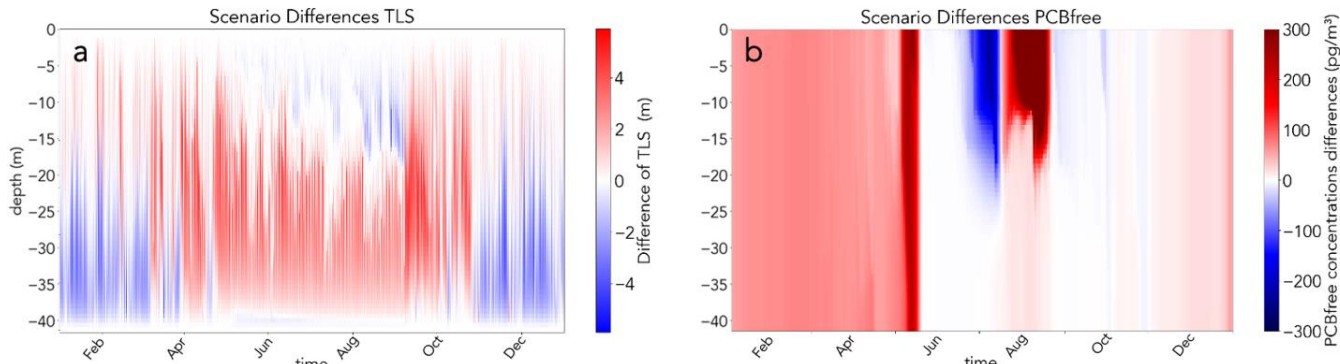

**Figure 14.** Results of simulations for: (**a1,a2,a3**) no-tide scenario, (**b1,b2,b3**) tide scenario (SNS, 2001 year). Concentrations of: (**a1,b1**) $PCB_{DOM}$ (pg m$^{-3}$); (**a2,b2**) freely dissolved $PCB_{153}$ (pg m$^{-3}$); (**a3,b3**) atmospheric flux of $PCB_{153}$ (pg m$^{-2}$ s$^{-1}$) (direction of flux—from atmosphere to water).

**Figure 15.** Differences between tide–no-tide scenarios for SNS region: (**a**) in turbulence length scale; (**b**) in $PCB_{free}$ concentration.

## 4. Summary

The fate of PCBs in the ocean is governed by regional hydrodynamic and biogeochemical regimes. PCB cycling in the four regions investigated in this study are governed by different processes.

In the northern part of the Baltic Sea, which is seasonally covered by sea ice, the atmospheric influx is blocked. Little primary production and turbulent mixing lead to PCBs being evenly distributed in this area with no significant seasonality.

In the deep Baltic basin, the water column is permanently stratified and characterized by an oxic surface and an anoxic deep layer. Starting in August/September, the atmospheric influx of PCB steadily increases in concentration in the lighter surface layer. With the onset of primary production around May, PCB is bound to organic matter and subsequently settles with organic particles. The model predicts an annual atmospheric influx of 320 PCB pg m$^{-3}$, which is transported to the sediment, most of it in a pronounced initial deposition flux in June (Figure 16a).

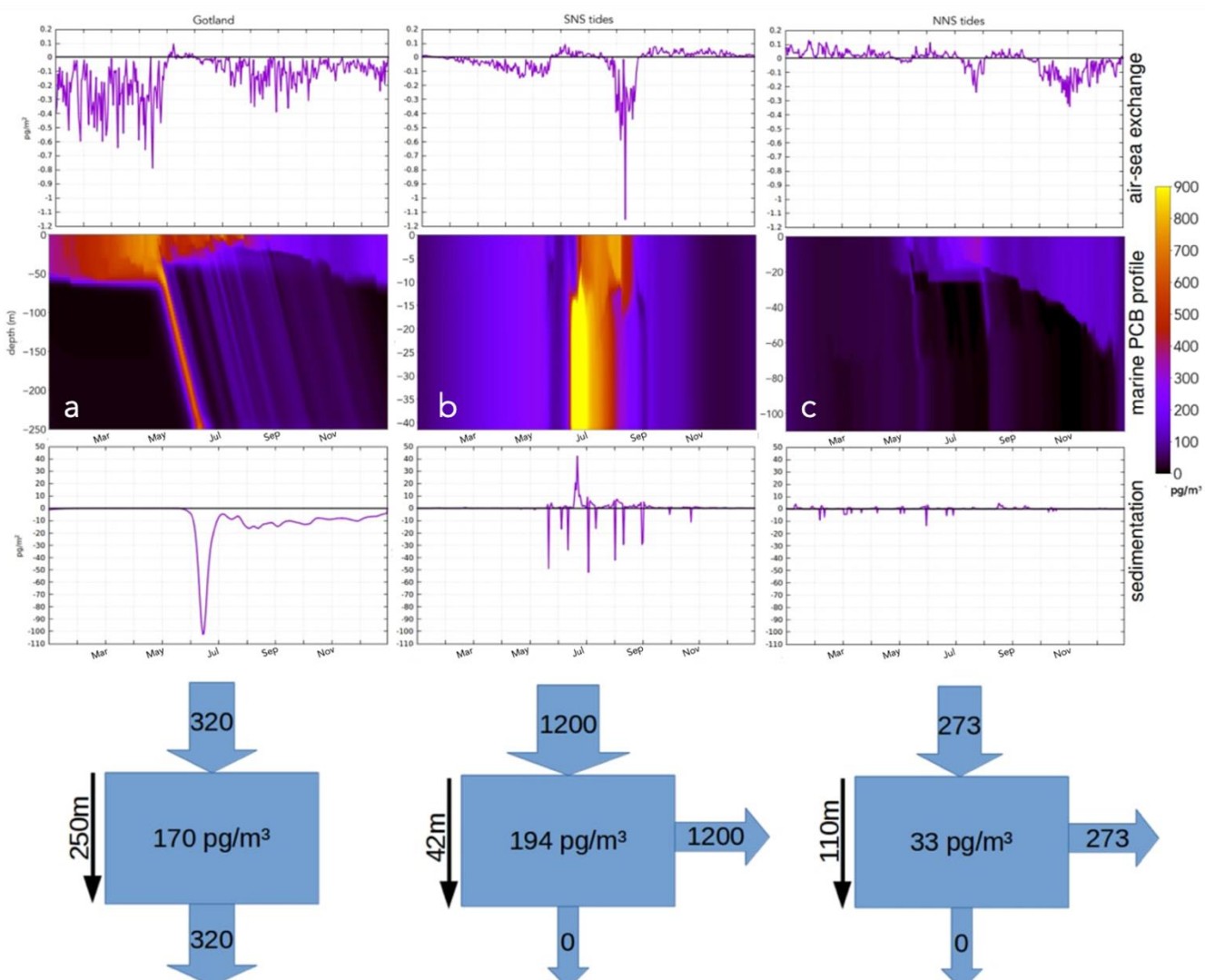

**Figure 16.** Summary picture of total concentrations and budgets for (**a**) Gotland Basin, (**b**) Southern North Sea (with tides) and (**c**) Northern North Sea (with tides) locations.

In the Northern North Sea, the water column is stratified during summer/autumn and mixed during winter/spring (Figure 16c). Due to high biological production, during summer, dissolved PCB is bound to dissolved and particulate organic carbon. The process

is similar to that described in the Gotland Deep. However, during winter, the mixed layer depth steadily increases, diluting PCBs in the surface layer over a larger volume. During this period, atmospheric PCBs are transported into the ocean. During spring, when the water column is mixed, any PCBs sedimented in the previous year are resuspended and evenly distributed over the water column until the next stratification and production cycle begins. On average, 273 pg m$^{-3}$ PCBs are transported from the atmosphere to the ocean. This PCB as well as any additional sources, such as rivers, are advected toward the Atlantic Ocean.

In the Southern North Sea, which is well-mixed throughout the year, PCBs are evenly distributed throughout the water column (Figure 16b). During late summer, there is almost no dissolved PCB left in the surface layer, which leads to a pronounced atmospheric influx during summer (1200 pg m$^{-3}$). During summer, we observe pronounced resuspension events, which can triple the total amount of PCB in the system temporarily. The full release of PCB from sediments back to the water column increases water concentrations and limits the ability of the water column regarding diffusive gaseous uptake from the atmosphere. This applies also for PCBs coming from rivers in this area.

## 5. Conclusions

We developed a marine biogeochemical model for polychlorinated biphenyls (PCBs). The model, which can resolve the fate of individual PCB congeners, was exemplarily evaluated for PCB$_{153}$. We used a 1D hydrodynamical host model to evaluate the fate PCB$_{153}$ under a variety of hydrophysical and biological conditions. We found that the two driving processes are primary production and vertical mixing. Firstly, high biological production binds dissolved PCBs to organic matter. The particulate OM then drove sedimentation, reducing the PCBs surface concentrations. The combination of reduced total PCBs and a smaller fraction of total PCB in the dissolved phase changed the concentration gradient between the atmosphere and ocean and therefore increased the atmospheric PCB flux into the ocean. Secondly, vertical mixing, which depends on tides, bathymetry, and wind regime, determined whether sedimentation is permanent, seasonal, or negligible. Regions of permanent sedimentation, such as the deep basins in the Baltic Sea, can be a major sink for PCBs, and thus are an important factor when determining long-range transport from the coast to remote regions. In regions with seasonal sedimentation, we found that, typically, PCBs are sedimented after the initial primary production peak and released in winter and spring. This leads to an enormous increase in bioavailable dissolved PCB concentrations right at the onset of early primary production. Therefore, it has a major impact on the bioaccumulation of PCBs. Here, we found that the underlying hydrodynamic characteristics strongly influence the fate and transport of PCBs in the water column, the benthic–pelagic coupling and exchange as well as the uptake to or release from the atmosphere. In the coastal ocean, pronounced tides have a strong impact on vertical mixing and the stability of the summer stratified surface layer. Thus, implementing realistic tides and interactive benthic–pelagic coupling is vital to correctly reproduce vertical mixing and thus the fate of PCB in shelf seas.

**Author Contributions:** Conceptualization, E.M. and C.S.; methodology, E.M.; software, J.B.; investigation, E.M.; writing—original draft preparation, E.M.; writing—review and editing, J.B. and C.S.; visualization, E.M. and J.B.; supervision, J.B. and C.S.; funding acquisition, C.S. All authors have read and agreed to the published version of the manuscript.

**Funding:** This work was partially funded by the H2020 project iGOSP under the ERA-PLANET program (Grant agreement no: 689443).

**Institutional Review Board Statement:** Not applicable.

**Informed Consent Statement:** Not applicable.

**Data Availability Statement:** Not applicable.

**Conflicts of Interest:** The authors declare no conflict of interest. The funders had no role in the design of the study; in the collection, analyses, or interpretation of data; in the writing of the manuscript, or in the decision to publish the results.

## Abbreviations

*List of abbreviations*

| | |
|---|---|
| **PCB** | polychlorinated biphenyls |
| **CB** | chlorinated biphenyls (biphenyls with one chlorine in the structure) |
| **POP** | Persistent organic pollutants |
| **SC** | Stockholm Convention |
| **SNS** | Southern North Sea |
| **NNS** | Northern North Sea |
| **GOTM** | 'General Ocean Turbulence Model' [29] |
| **ECOSMO** | 'ECOSystem MOdel', is a 3D fully coupled physical–biogeochemical model [22] |
| **FABM** | The Framework for Aquatic Biogeochemical Models [28] |
| **NPZD** (model) | nutrients, phytoplankton, zooplankton, detritus ecosystem model |
| **CTD** | oceanography instrument used to measure the electrical conductivity, temperature, and pressure of seawater |
| **WOA** | World Ocean Atlas |
| **EMEP** | 'European Monitoring and Evaluation Programme' [26] |
| **LRT** | long-range transport |
| **OM** | organic matter |
| **POM** | particulate organic matter |
| **DOM** | dissolved organic matter |
| **SSE** | sea surface elevation |
| **ORCM** | oxidative ring cleavage mechanism |

*List of variables and parameters used in this article*

| | |
|---|---|
| $PCB_{free}$ | $PCB_{153}$ concentration in the dissolved phase [pg·m$^{-3}$] |
| $PCB_{POM}$ | concentration of $PCB_{153}$ on POM [pg·m$^{-3}$] |
| $PCB_{DOM}$ | concentration of $PCB_{153}$ on DOM [pg·m$^{-3}$] |
| $PCB_{sed}$ | concentration of $PCB_{153}$ in sediment [pg·m$^{-2}$] |
| $POM$ | particulate organic matter concentration [mgC·m$^{-3}$] |
| $DOM$ | dissolved organic matter concentration [mgC·m$^{-3}$] |
| $K_{ow}$ | octanol–water coefficient |
| $K_{oc}$ | organic carbon–water partitioning coefficient |
| $k_{sorp(1,2)}$ | rate constant of $PCB_{153}$ sorption on POM and DOM respectively [s$^{-1}$] |
| $k_{desorp(1,2)}$ | rate constant of $PCB_{153}$ desorption from POM and DOM respectively [s$^{-1}$] |
| $k_{rem1}$ | rate constant of detritus (POM) remineralization [s$^{-1}$] |
| $k_{rem2}$ | rate constant of DOM remineralization [s$^{-1}$] |
| $k_{rem3}$ | rate constant of sediment remineralization [s$^{-1}$] |
| $k_{resusp}$ | resuspension rate (bottom shear stress depended) [s$^{-1}$] |
| $k_{sed}$ | sedimentation rate of $PCB_{POM}$ [s$^{-1}$] |
| $k_{bur}$ | burial rate of $PCB_{153}$ [s$^{-1}$] |
| $k_{photo}$ | rate constant of $PCB_{153}$ photolytic degradation [s$^{-1}$] |
| $k_{bio}$ | rate constant of $PCB_{153}$ biological degradation [s$^{-1}$] |
| $K_{bio}$ | parametrization coefficient for $k_{bio}$ calculation |
| $k_{diff(1,2)}$ | rates of $PCB_{153}$ diffusive exchange with dissolved in water pollutants [s$^{-1}$] |
| $k_{deg}$ | rate constant of pollutant degradation in sediments [s$^{-1}$] |
| $Q_{153}$ | quantum yield of photolytic degradation |
| $F_{PCB153}$ | photon flux [μmol·m$^{-2}$·s$^{-1}$ = μE] |
| $\lambda$ | wavelength [m]; |
| $I$ | sunlight irradiance in UV range of spectrum [W·m$^{-2}$] |
| $\varepsilon_{153}$ | extinction coefficient for $PCB_{153}$ [m$^2$·mol$^{-1}$] |
| $EXT_{TOT}$ | total extinction, includes marine water, DOM, detritus and phytoplankton [m$^2$·mol$^{-1}$] |
| $k_a$ | mass transfer coefficient in air [m·s$^{-1}$] |
| $K_w$ | mass transfer coefficient in water [m·s$^{-1}$] |

| | |
|---|---|
| $C_a$ | PCB$_{153}$ concentration in air [pg·m$^{-3}$] |
| $\Delta H$ | enthalpy of dissolution [J·mol$^{-1}$] |
| $\Delta S$ | entropy of dissolution [J·mol$^{-1}$·K$^{-1}$] |
| $R$ | gas constant [J·K$^{-1}$·mol$^{-1}$] |
| $u_{10}$ | wind speed at 10 m above a water [m·s$^{-1}$] |
| $D_{wi}, D_{ai}$ | diffusivities of PCB$_{153}$ in the air and water [cm$^2$·s$^{-1}$] |
| $D_{wCO2}$ | CO$_2$ diffusivities in the water [cm$^2$·s$^{-1}$] |
| $D_{aH2O}$ | H$_2$O diffusivities in the air [cm$^2$·s$^{-1}$] |
| $MW_{air}, MW_{PCB}$ | molar weight of an air and chosen PCB [g·mol$^{-1}$] |
| $V_{air}, V_{PCB}$ | molar volume of an air and chosen PCB [cm$^3$·mol$^{-1}$] |
| $P$ | air pressure [atm] |
| $\nu_{H2O}$ | kinematic viscosity (at 25 °C) [cm$^2$·s$^{-1}$] |
| $F_{ow}$ | open water flux [pg·m$^{-2}$·s$^{-1}$] |
| $F$ | atmospheric flux [pg·m$^{-2}$·s$^{-1}$] |
| $A_i$ | sea ice compactness (the part of the cell covered by sea ice) |
| $temp_c$ | temperature control factor [°C$^{-1}$] |
| $u_*^b$ | friction velocity |
| $u_{cr}^b$ | critical velocity |
| $U_1, V_1$ | components of mean velocity at the center of the lowest cell |

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
