# Peer review of "Hydrodynamic Impacts on the Fate of Polychlorinated Biphenyl 153 in the Marine Environment"

_water, doi:10.3390/w14233952_

Round 1

Reviewer 1 Report

Reviewers' comments:

This is a review of the manuscript entitled “Hydrodynamic impacts on the fate of PCB153 in the marine environment” (water-1954255) by Elena Mikheeva. In this manuscript, the authors developed a new chemical model, based on GOTM-ECOSMO-FABM model framework, and exemplarily present results for PCB153 based on 1D simulations of 4 regions in the North-Baltic Sea. This is of significant value and significance for understanding the status of pollution in the marine environment. Therefore, it is suitable for publication in this journal. However, there are some minor problems in this manuscript.

Minor problems:

1.       Title: PCB153 in the title gives the full name.

2.       Do not have numbers in the abstract.

3.       The paper version is 2020, and it is recommended to use the latest version.

4.       The references of this paper are too old, so it is suggested to add the latest achievements.

5.       It is suggested to add the validation results and applicability of the model.

Author Response

Dear Reviewer #1,
We want to thank you for your insightful comments and the effort put into
improving this article. In the following we address all issues raised by the re-
viewers:
Comments 1 - 3:
1. Title: PCB153 in the title give the full name.
2. Do not use numbers in the abstract.
3. The paper version is 2020, and it is recommended to use the latest
version.
A: We would like to acknowledge Reviewer #1 for the clear and concise
comments. The first three were of technical nature and we changed the
original manuscript accordingly.
In particular, we have changed the abbreviation of PCB153 in the title to
the full name of pollutant and removed numbers from the abstract. Also,
we renewed the paper version from 2020 to 2022 and added lines for easier
communication between reviewers and authors.
Comment 4: The references of this paper are too old, so it is suggested to
add the latest achievements.
A: We added/updated following citations to the manuscript:
2. Wolska, L., Mechli ́nska, A., Rogowska, J. and Namie ́snik. J. (2012)
Sources and Fate of PAHs and PCBs in the Marine Environment, Crit.
Rev. Environ. Sci. Technol., 42(11), 1172–1189, https://doi.org/10.1080/10643389.2011.556546
(line 865)
4. Schwarzenbach, R. P., Gschwend, P. M., and Imboden, D. M. (2016).
Environmental organic chemistry (3rd ed.), Wiley, https://doi.org/10.1002/0471649643
(line 870)
10. Zhou, W., Zhai, Z., Wang, Z. and Wang, L. (2005). Estimation of
n-octanol/water partition coefficients (KOW) of all PCB congeners by density functional theory. J. Mol. Struct.: THEOCHEM., 755, 137–145
https://doi.org/10.1016/j.theochem.2005.08.020.(line 883)
19. Kong, D., MacLeod, M. and Cousins, I. (2014). Modelling the influ-
ence of climate change on the chemical concentrations in the Baltic Sea
region with the POPCYCLING-Baltic model. Chemosphere, 110, 31–40.
https://doi.org/10.1016/j.chemosphere.2014.02.044. (line 902)
20. Lamon, L., MacLeod, M., Marcomini, A. and Hungerb ̈uhler, K.
(2012). Modeling the influence of climate change on the mass balance of
polychlorinated biphenyls in the Adriatic Sea. Chemosphere, 87, 1045–51.
https://doi.org/10.1016/j.chemosphere.2012.02.010. (line 905)
23. McLachlan, M., Undeman, E., Zhao, F. and MacLeod, M. (2018).
Predicting global scale exposure of humans to PCB 153 from historical
emissions. Env. Sci.: Processes Impacts. 20, 747–756, https://doi.org/20.
10.1039/C8EM00023A (line 912)
Comment 5: It is suggested to add the validation results and applicability
of the model.
A: Due to lack of observational data of freely dissolved PCB153 in chosen
areas, we can only state that our values are within a realistic range. Our
study was aimed at investigating and quantifying processes, driving marine
PCBs cycling. The 1D model presented in this study allowed us to do
exactly that: to compare hydrodynamic conditions on patterns of PCB153
distribution and speciation.

Reviewer 2 Report

This manuscript presented an interesting study of the fate and transport simulation and hydrodynamic impacts of PCB153 in the marine environment. The subject matter falls well within the scope of the journal. The authors proposed a 1-D chemical modeling approach based on the GOTM-ECOSMO-FABM model framework. And the proposed model is further adopted to simulate PCB153 fate conditions in four regions of the North and Baltic Seas. However, the manuscript needs to highlight innovative and unique contributions to the model development. The methodology section should be more logical and concise and highlight the comparisons with other PCBs chemical models to evaluate the performance and improvement of the proposed model. Information on modeling calibration and validation should also be stated in the manuscript. Below are some specific comments to improve the quality of the manuscript:

·        The keywords are missing. It is better to give the line numbers.

·        Are there models developed by other papers using the same or similar chemical model frameworks (e.g., ECOSMO)?

·        What are the improvements or advantages of the proposed PCB cycling numerical model? What are “new” points of the proposed model? Why can it be called a “new chemical model?

·        Can the proposed model address the challenges or limitations of box models or 3D models mentioned in the introduction? The modeling innovations are not addressed clearly in the introduction.

·        Section 2.1, what does “new coupled 1D model” mean? What is coupled?

·        Figure 2, the meaning of symbols here is unclear. An explanation is required.

·        Full names of abbreviations (e.g., CTD, SD) need to be provided.

·        Equation 4, how to define the value of Alpha DOM?

·        Table 2, is there any evidence or reference that helps define the estimates?

·        Some parameters are not clearly defined. For example, what is the unit of T in Equation 13? What is the value of KBIO? Does T in Equation 9 also represent temperature? What temperature is used in the model? Water temperature or air temperature? The definition and illustration of equations should be checked.

·        The manuscript contains too many similar parameters (e.g., PCB, PCBDOM, PCBfree, PCBsed, PCB153, Ksorp1, Ksorp2, Kdesorp1, Kdesorp2), a list of abbreviations should be provided to help understand the methodology.

·         Section 2.6, the parameter values used in the North Sea and the Baltic Sea regions should be provided to help evaluate the performance of the model.

·        How to balance the bias and compatibility brought by equations from different sources? Was there any calibration or validation done before using the model for the case?

·        What are the differences in the characteristics of the North Sea and Baltic Sea regions, and why are these regions chosen?

·        Figure 6, what do the letters (e.g., M, M, J, S) on the horizontal axis (time) mean?

·        Some relevant references are suggested: 10.1021/es071432d; 10.1080/10643389.2011.556546; 10.1016/j.jhazmat.2022.129260; 10.3389/fmars.2021.768715; 10.1016/j.envpol.2013.06.028

·        There are some grammatical mistakes. The language should be thoroughly checked and polished.

Author Response

Dear Reviewers,
We want to thank you for your insightful comments and the effort put into
improving this article. In the following we address all issues raised by the re-
viewers:
Suggestions and comments from Reviewer #2 were taken into considera-
tion and the manuscript has been adjusted accordingly.
Comment 1: The keywords are missing. It is better to give the line num-
bers.
A: We have added a line with keywords and line numbers to manuscript.
Keywords: PCB; POPs; North and Baltic Sea; 1D model; pollutant mod-
eling (line 32)
Comment 2: Are there models developed by other papers using the same
or similar chemical model frameworks (e.g., ECOSMO)?
A: GOTM is a widely used 1d ocean model. The ecosystem model ECOSMO
is used by a number of researchers. We also developed a model for Hg
transport based on ECOSMO.
Comment 3: What are the improvements or advantages of the proposed
PCB cycling numerical model? What are “new” points of the proposed
model? Why can it be called a “new chemical model?
A: We developed a comprehensive marine PCB153 model representing all
relevant chemical and physical processes influencing PCB153 cycling. We considered specific properties of the pollutant, related to their chemical na-
ture, such as kinetic sorption/desorption to/from OM, diffusive exchange
with atmosphere and sediment, and biological and photolytic degradation.
Comment 4: Can the proposed model address the challenges or limitations
of box models or 3D models mentioned in the introduction? The modeling
innovations are not addressed clearly in the introduction.
A: Here we present a new numerical model for PCB cycling in the marine
environment, that considers all biogeochemical and physical processes in
the water column, the benthic-pelagic coupling along with chemical fea-
tures of PCB congener. With this model we investigate different processes
in the water column and their influence on the behavior of one PCB con-
gener, considering its specific physico-chemical properties, here – PCB153.
(lines 121-125)
Comment 5: Section 2.1, what does “new coupled 1D model” mean? What
is coupled?
A: For this study, we use coupled model system of hydrodynamic host
model GOTM, marine ecosystem model ECOSMO and couple the PCB
chemistry module by using Framework for Aquatic Biogeochemical Models
(FABM) (lines 152-156)
Comment 6: Figure 2, the meaning of symbols here is unclear. An expla-
nation is required
A: Added descriptions of all model parameters in Fig. 2. They can be
found on line 162, with detailed description in the Section 2.3.
Comments 7, 11
Comment 7: Full names of abbreviations (e.g., CTD, SD) need to be
provided.
Comment 11: The manuscript contains too many similar parameters (e.g.,
PCB, PCBDOM, PCBfree, PCBsed, PCB153, Ksorp1, Ksorp2, Kdesorp1,
Kdesorp2), a list of abbreviations should be provided to help understand
the methodology
A: we have provided detailed list of abbreviations, parameters and vari-
ables, used in the manuscript. This list can be found at line 777.
Comment 8: Equation 4, how to define the value of Alpha DOM?
A:Alpha DOM in eq. 4,5 represents part of the mortality component of
phytoplankton and zooplankton (MORTBIO) turning to DOM. (line 240)
Comment 10: Some parameters are not clearly defined. For example,
what is the unit of T in Equation 13? What is the value of KBIO? Does
T in Equation 9 also represent temperature? What temperature is used
in the model? Water temperature or air temperature? The definition and
illustration of equations should be checked.
A: We redefined all parameters which were unclear such as units for tem-
perature (degree celcius) (line 292, 304, 385). In addition, we would like to clarify that temperature values in all equations are water temperatures,
not air. These data were prescribed from World Ocean Atlas (WOA) and
mentioned in Section 2.2.
Comments 12, 14
Comment 12: Section 2.6, the parameter values used in the North Sea and
the Baltic Sea regions should be provided to help evaluate the performance
of the model Comment 14: What are the differences in the characteristics
of the North Sea and Baltic Sea regions, and why are these regions chosen?
A: We kindly wanted to point, that parameter values used in the North
Sea and the Baltic Sea regions were provided in Section 2.2 with detailed
description of GOTM. However, we have added the reference to this chap-
ter into Section 2.6 (line 487). We think this chapter provides information
about reasons for choosing regions, with detailed explanation about hy-
drodynamic differences. For better linkage between different parts of this
article we added reference to this Chapter 2.6 into introduction (line 129).
Comment 13: How to balance the bias and compatibility brought by equa-
tions from different sources? Was there any calibration or validation done
before using the model for the case?
A: Due to lack of observational data of freely dissolved PCB153 in chosen
areas, we can only state that our values are within a realistic range. Our
study was aimed at investigating and quantifying processes, driving marine
PCBs cycling. The 1D model presented in this study allowed us to do
exactly that: to compare hydrodynamic conditions on patterns of PCB153
distribution and speciation.
Comment 15: Figure 6, what do the letters (e.g., M, M, J, S) on the
horizontal axis (time) mean?
A: We exchanged x-axis months names in Figs. 6-16 from letters to 3
letter month abbreviation.
Comment 16: Some relevant references are suggested: 10.1021/es071432d;
10.1080/10643389.2011.556546; 10.1016/j.jhazmat.2022.129260; 10.3389/fmars.2021.768715;
10.1016/j.envpol.2013.06.028
A: We would like to give special thanks for the suggested relevant ref-
erences. We used paper 10.1021/es071432d as a new reference (lines
43,44,865).
Comment 17: There are some grammatical mistakes. The language should
be thoroughly checked and polished.
A: We have checked the current manuscript for grammatical mistakes.
Also, we have polished the language in line 152.